# RGMIL: Guide Your Multiple-Instance Learning Model with Regressor

Zhaolong Du[1], Shasha Mao[1]*, Yimeng Zhang[1], Shuiping Gou[1], Licheng Jiao[1], Lin Xiong[2]
[1]Xidian University, [2]SenseTime.

## Abstract

In video analysis, an important challenge is insufficient annotated data due to the rare occurrence of the critical patterns, and we need to provide discriminative frame-level representation with limited annotation in some applications. Multiple Instance Learning (MIL) is suitable for this scenario. However, many MIL models paid attention to analyzing the relationships between instance representations and aggregating them, but neglecting the critical information from the MIL problem itself, which causes difficultly achieving ideal instance-level performance compared with the supervised model. To address this issue, we propose the ***Regressor-Guided MIL network*** (**RGMIL**), which effectively produces discriminative instance-level representations in a multi-classification scenario. In the proposed method, we make full use of the *regressor* through our newly introduced *aggregator*, ***Regressor-Guided Pooling*** (**RGP**). RGP focuses on simulating the correct inference process of humans while facing similar problems without introducing new parameters, and the MIL problem can be accurately described through the critical information from the *regressor* in our method. In experiments, RGP shows dominance on more than 20 MIL benchmark datasets, with the average bag-level classification accuracy close to 1. We also perform a series of comprehensive experiments on the MMNIST dataset. Experimental results illustrate that our *aggregator* outperforms existing methods under different challenging circumstances. Instance-level predictions are even possible under the guidance of RGP information table in a long sequence. RGMIL also presents comparable instance-level performance with S-O-T-A supervised models in complicated applications. Statistical results demonstrate the assumption that a MIL model can compete with a supervised model at the instance level, as long as a structure that accurately describes the MIL problem is provided. The codes are available on `https://github.com/LMBDA-design/RGMIL`.

## 1 Introduction

Multiple-instance learning (MIL) [3][33] is commonly used for the binary classification tasks in which instances lack individual labels and are grouped into bags. In MIL, a bag is labeled as positive if it contains at least one positive instance, otherwise, it is negative. Its goal is to learn the patterns that characterize positive bags. The models of MIL have been widely applied in various applications, such as anomaly detection [16, 13], pathology diagnosis [24, 4], etc.

At present, many deep MIL models adopt *attention-based modules* [12] to solve the MIL problems, which aggregate instance-level representations in various ways. Such approaches work during the stage of aggregating instance-level representations and perform well when the given instance-level representations are with good discriminability. However, in some applications, such as facial pain estimation, we face a scenario where there are no discriminative instance-level representations or the instance-level representations need to be explicitly learned. For them, simply learning the aggregation

---

*Corresponding author. `email:ssmao@xidian.edu.cn`

37th Conference on Neural Information Processing Systems (NeurIPS 2023).

process is far from enough when the instance-level performance is important. Most of the current methods fail to achieve satisfying instance-level performance in this case.

To address this issue, we propose a new model of multiple-instance learning based on the guidance of regressor, named as *Regressor-Guided MIL network* (RGMIL), which achieves promising instance-level performance. Compared with existing methods, our contributions are listed as follows:

- RGMIL can address the general multi-class classification problem in MIL scenarios by learning discriminative instance-level features. RGMIL accurately describes the MIL problem through the newly introduced aggregator, Regressor-Guided Pooling (RGP). With a accurate description of the MIL problem, we can fully transfer the learning process into the direct learning of instance-level representations, without the requirement for increasingly complicated aggregation processes.

- RGMIL demonstrates outstanding performance and expected functionality on public datasets, showcasing the essential role of RGP. Indicators show that RGP performs well under different challenging circumstances, and instance-level predictions are even possible with the guidance of RGP information in a long sequence. Experimental results validate that the regressor guidance possibly brings similar instance-level performance to a supervised model.

- Excepted the simulated data (MMNIST), the effectiveness of RGMIL is also validated on a real pain estimation dataset (UNBC). Specifically, *from the perspective of pain estimation, RGMIL can be as effective as supervised learning models.*

## 2 Related Works

### 2.1 The General Models of MIL

In the classical case of Multi-Instance Learning (MIL), a binary-classification problem is built and solved based on bags, and each bag is formed by several instances. In general, a bag is defined as $\mathbf{X} = [\mathbf{x}_1, \mathbf{x}_2, \ldots, \mathbf{x}_t]$, and its label is formulated by

$$\mathbf{Y} = \max_k \{\mathbf{y}_k\}, \ \ k \in [1, t]. \tag{1}$$

Each bag includes $t$ instances ($\mathbf{X}_i \in \mathbb{R}^{D \times t}$) and one label ($\mathbf{Y}_i \in \{0, 1\}$). The task of MIL is to recognize what type of instances-level representation in the bag $\mathbf{X}_i$ could make the bag label $\mathbf{Y}_i$ be one (positive). Classical MIL also assumes that there are neither ordering nor dependency of instances within a bag. There are two main architectures in the frameworks of MIL, shown as follows.

*Architecture 1*: *Instance-level Backbone + Regressor + Aggregator*

In this architecture, the MIL model generally takes an *instance-level backbone* to obtain the instance-level representations ($\mathbf{fs} = [\mathbf{f}_1, ..., \mathbf{f}_t]$) which involves each instance in a bag ($\mathbf{X} = [\mathbf{x}_1, .., \mathbf{x}_t]$), where $\mathbf{f}_i$ is corresponding to $\mathbf{x}_i$, $\mathbf{f}_i \in \mathbb{R}^c$, $i \in [1, t]$. Then, the instance-level representations ($\mathbf{fs}$) are fed into the *regressor* to predict the score or possibility ($\mathbf{p}_i$) of categories for each instance. Finally, the *aggregator* is used to aggregate the predicted scores ($\mathbf{P} = [\mathbf{p}_1, ..., \mathbf{p}_t]$) and obtain the final bag-score $\mathbf{p_{bag}}$. Suppose $\rho$ express the aggregation function, the aggregator is formulated as $\mathbf{p_{bag}} = \rho(\mathbf{P})$, and $\rho$ should follow the *permutation-invariant restriction*:

$$\rho(\mathbf{P}) = \rho(\mathbf{PT}), \tag{2}$$

where $\mathbf{T}$ expresses a arbitrary permutation matrix with appropriate dimension. In this architecture, the *max pooling* and *average pooling* are two most widely used aggregators. The max pooling (MXP) takes the highest value in $\mathbf{P}$ to produce $\mathbf{p_{bag}}$, while the average pooling takes the average value in $\mathbf{P}$, and both of them comply with Equation (2).

*Architecture 2.*: *Instance-level Backbone + Aggregator + Regressor*

In this architecture, the MIL model produces the bag-level representation ($\mathbf{F}$) directly via a *backbone+aggregator* structure. For example, in a image classification problem, a 2D-CNN backbone is used to get instance-level representation matrix $\mathbf{fs} \in \mathbb{R}^{c \times t}$ and then aggregator $\rho$ is applied to get the bag-level representation $\mathbf{F} = \rho(\mathbf{fs})$. The function $\rho$ still satisfy Equation (2). Then, each bag is predicted by the *regressor* based on the bag-level representation $\mathbf{F} \in \mathbb{R}^c$. In this architecture, the most widely used aggregators are *attention-based pooling (ABP)* [12], *gated attention-based pooling*

*(G-ABP)* [12], and their variations [15][30][32]. Compared with *Architecture 1*, Wang et al.[27] advocates *Architecture 2* which produce the bag-level representation $\mathbf{F}$, since they often perform better in terms of the bag-level classification.

## 2.2 Attention-based Pooling Series

At present, Attention-Based Pooling (ABP) and its variations are widely adopted in many mainstream methods of MIL. Generally, ABP is formulated as

$$\mathbf{F} = \sum_{k=1}^{t} a_k \mathbf{f}_k, \quad a_k = \frac{\exp\left\{\mathbf{w}^\top \tanh\left(\mathbf{V}\mathbf{f}_k^\top\right)\right\}}{\sum_{j=1}^{t} \exp\left\{\mathbf{w}^\top \tanh\left(\mathbf{V}\mathbf{f}_j^\top\right)\right\}} \tag{3}$$

where $\mathbf{w} \in \mathbb{R}^t$ and $\mathbf{V} \in \mathbb{R}^{t \times c}$ are both parameters to be trained. $\mathbf{f}_k \in \mathbb{R}^c$ is the $k$th instance representation in a bag. ABP provides the bag representation by weighted sum of representations of different instances. Moreover, many variations of ABP has been proposed, such as Dual-Stream MIL(2021) [15], DTFD-MIL(2022) [30], GAMIL(2023) [32]. Dual-Stream MIL [15] (DSP) used MXP and an MXP-guided ABP as two streams to aggregate representations and obtained the S-O-T-A level performance on MIL benchmarks. DTFD-MIL [30] divided a bag into several pseudo-bags when the bag size was really large and applied ABP on every pseudo-bag twice in a two-stage manner. They still follow the main idea of ABP: *weights should rely on representations*.

In many applications, such as the whole slide image based pathology diagnosis [9], there is no strong requirement for sufficiently discriminative instance-level representations. The model is provided with bags with the bag length $t$ over large, and is asked to make accurate bag-level predictions. Most of the current models like ABP variations introduced above would only *focus in learning of aggregation process* or *limit the analysis process between instances* given the instance-level representations fixed.

## 2.3 Pain or No Pain

In some real-world applications, it is excepted to obtain the accurate instance-level prediction or representation, such as pain estimation. Most studies of pain estimation based on facial expression [6],[7],[18] rely on the Facial Action Coding System (FACS) [5], which is a rating system that distinguishes 44 facial movements, called Action Units (AU). Painful expressions are associated with some of these AUs. Earlier, Prkachin et al. [19] developed the Prkachin and Solomon Pain Intensity (PSPI) index based on these AUs, which is a 17-point scale reflecting the degree of pain, where 0 indicates no pain and 16 indicates extreme pain. Whereas, due to the imbalanced distribution of the PSPI index, researchers redefined the pain intensity based on the PSPI index. In general, the pain intensity are divided into four categories based PSPI: no pain (0), mild pain (1 to 2), severe pain (3 to 5), and extreme pain (the above 5). At present, there are some existing MIL algorithms, such as MI-DORF[22], MIR[31], but it is difficult for a pure MIL algorithm to provide ideal instance-level performance when compared with supervised deep models.

# 3 The Proposed Method

## 3.1 Assumptions

In this paper, we design the model (RGML) with the guidance of two assumptions shown as follows.

**Assumption 1** *It is realistic for a MIL model to achieve the performance similar to a supervised model at the instance level, as long as it has a well-designed architecture.*

When facing MIL problems, we start from the *Architecture 2* given in section 2.1. Compared to the fully supervised model, a new component *aggregator* is added. Here we would not focus on whether the data is in the form of bags or instances. From the perspective of bags, it is still fully supervised. But, *it is simply another model with an additional component, and the problem to be solved is not even ambiguous*. When a MIL problem is treated in this way, there is reason to make this assumption.

**Assumption 2** *Performance bottleneck of the current MIL model is the aggregator function $\rho$.*

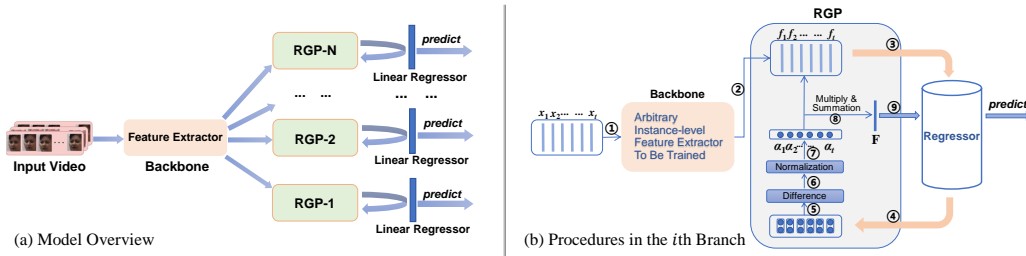

Figure 1: Overview of Regressor-Guided multiple-instance Learning Network (RGMIL)

Compared to the supervised learning, the available information obtained by MIL is limited. Therefore, the instance-level performance limit of MIL should be equivalent to the fully supervised mode. Two main architectures introduced above differ from the fully supervised ones only on the presence of the aggregator. We argue that the inability of MIL to achieve the performance of the fully supervised mode is due to the limitation of the learning stage of the new component (aggregator).

So what makes a good aggregator? We assume that if we could describe the key differences between a MIL problem and a supervised problem in $\rho$ well enough, we could get an accurate MIL model. Recall what CNN does. Early in computer vision, representations were provided as fixed to *train a pure regressor*. CNN took a step forward by its *accurate description*, which makes it possible to train the *backbone+regressor* at the same time, and get even better performance. The main task of the new component $\rho$ would also be to *provide accurate description*.

### 3.2 Introduction of RGMIL

RGMIL is proposed for a multi-classification scenario, as shown in Fig.1. It is achieved by $N$ branches, each of which learns to distinguish one corresponding *non-negative class (with non-zero label)*. Thus $N$-branch RGMIL is suitable to solve an $(N + 1)$-class classification problem. To solve the pain estimation problem, it requires only a sequence of frames and the highest pain intensity level ($0$ to $N$, discrete) of the sequence as input.

During each training step, a bag composed of $t$ images is fed into the model, denoted as $\mathbf{X} = [\mathbf{x}_1, \mathbf{x}_2, \ldots, \mathbf{x}_t], \mathbf{X} \in \mathbb{R}^{D \times t}$, with a bag label $\mathbf{Y_{bag}} \in [0, N]$ which indicates ***the max instance label*** in the bag. We convert $\mathbf{Y_{bag}}$ to a special vector $\mathbf{Y}$, similar to but not the same as one-hot encoding:

$$\mathbf{Y} \in \mathbb{R}^N; \mathbf{Y_i} = \begin{cases} 1, & \text{iff } \mathbf{Y_{bag}} = \mathbf{i} \\ 0, & \text{otherwise} \end{cases} \ (\mathbf{i} \in [1, N]). \tag{4}$$

Then, the backbone will produce the instance-level representations $\mathbf{fs} = [\mathbf{f}_1, \mathbf{f}_2, \ldots, \mathbf{f}_t] \in \mathbb{R}^{c \times t}$ corresponding to each image in this bag. The representations $\mathbf{fs}$ are fed into all the branches at the same time. For each branch $\mathbf{B^i}$, we have a corresponding linear regressor with parameters $\mathbf{W_i} \in \mathbb{R}^{c \times 2}$ and $\mathbf{b_i} \in \mathbb{R}^2(\mathbf{i} \in [1, N])$. Parameters in both the RGP and the regressor will be **NOT** involved in the gradient descent. To be clear, since our goal is to produce discriminative instance-level representations, only the backbone component (instance-level feature extractor) needs to be trained. For the RGP and regressor components, they are just a means of describing the MIL problem. They describe the MIL problem through specific structures, and as long as the structure is **clear enough to provide an accurate description**, it would be sufficient.

In the branch $\mathbf{B^i}$, given the instance-level representation matrix $\mathbf{fs} \in \mathbb{R}^{c \times t}$, the RGP works as follows:

$$\mathbf{H} = \mathbf{fs}^\top; \mathbf{P}_k = \mathbf{W_i}^\top \mathbf{H}_k + \mathbf{b_i}; (k \in [1, t], \mathbf{P} \in \mathbb{R}^{t \times 2}) \tag{5}$$

$$\widehat{P} = \mathbf{P}^\top; \mathbf{p} = \widehat{P}_2 - \widehat{P}_1; (\mathbf{p} \in \mathbb{R}^t) \tag{6}$$

$$\mathbf{w} = \frac{\mathbf{p} - E[\mathbf{p}]}{\sqrt{\mathrm{Var}(\mathbf{p}) + \epsilon}}; (\mathbf{w} \in \mathbb{R}^t, \epsilon \to 0) \tag{7}$$

$$\vec{\alpha} \in \mathbb{R}^t; \vec{\alpha}_k = \frac{\exp(\mathbf{w}_k)}{\sum_{j=1}^t \exp(\mathbf{w}_j)}; (\vec{\alpha}_k \in (0, 1)) \tag{8}$$

$$\mathbf{F_i} = \sum_{k=1}^{t} \vec{\alpha}_k \mathbf{H}_k; \, (\mathbf{F_i} \in \mathbb{R}^c) \tag{9}$$

Instance classification logits are provided in Equation 5. Difference values are then calculated to indicate the importance of instances in Equation 6. Normalization like Equation 7 is adopted to prevent the gradient vanishing in Equation 8. Use Equation 8 and 9 to get the final bag representation $\mathbf{F_i}$. In a MIL problem, we obey the inference rule that **if there are positive instance-level patterns, the bag is positive**. That is to say, there is *consistency between the discrimination* of critical instance-level patterns and the bag representation. We argue that this is a key element to describe a MIL problem, and we use a *shared regressor* to express this consistency between them. In summary, the RGP *asks for the current result of the regressor*, and always tries to aggregate the instance representations *based on the current judgments made by the regressor*, then dynamically adjusts current judgments in the training process —— Much like what humans would do.

Return to the process of forward propagation. For the branch $\mathbf{B^i}$, given the bag representation $\mathbf{F_i} \in \mathbb{R}^c$, the corresponding regressor parameters will be ***used again*** to produce the prediction $Z^{\mathbf{i}} \in \mathbb{R}^2$, which will be used to calculate the loss $\mathbf{L^i}$ in this branch($\mathbf{i} \in [1, N]$), formulated as follows:

$$Z^{\mathbf{i}} = \mathbf{W_i}^\top \mathbf{F_i} + \mathbf{b_i}; \, (Z^{\mathbf{i}} \in \mathbb{R}^2) \tag{10}$$

$$\mathbf{L} = \sum_{\mathbf{i=1}}^{\mathbf{N}} \mathbf{L^i}; \text{ where } \mathbf{L^i} = \begin{cases} -\log\left\{ \frac{\exp(Z_{1+\mathbf{Y_i}}^{\mathbf{i}})}{\exp(Z_1^{\mathbf{i}}) + \exp(Z_2^{\mathbf{i}})} \right\} & \text{iff } \mathbf{i} \geqslant \mathbf{Y_{bag}} \\ 0 & \text{otherwise} \end{cases} \tag{11}$$

The loss value $\mathbf{L^i}$ on the corresponding branch $\mathbf{B^i}$ is calculated as shown above. For a MIL problem, the bag label $\mathbf{Y_{bag}}$ is the maximum instance label in the bag. Based on this, we give the vector $\mathbf{Y}$ that describes the presence of instances with different labels in this bag. $\mathbf{Y_i} = 0$ means no instance with label $\mathbf{i}$ presented in this bag, while $\mathbf{Y_i} = 1$ means presence. Since $\mathbf{Y_{bag}}$ is *the maximum instance label*, some bits of $\mathbf{Y}$ would be *reliable*. For $\mathbf{Y_i}$ where $\mathbf{i} \in [\mathbf{Y_{bag}}, \mathbf{N}]$, it gives the correct ground truth, while for $\mathbf{Y_i}$ where $i \in [\mathbf{1}, \mathbf{Y_{bag}} - \mathbf{1}]$ it is not reliable. So for the branch $\mathbf{B^i}$ where $i \in [\mathbf{1}, \mathbf{Y_{bag}} - \mathbf{1}]$, it is meaningless to calculate the loss since it may contain false information. We set the loss on these branches to 0, as shown in Equation 11. For other *reliable branches*, we calculate the cross-entropy loss respectively. The total loss $\mathbf{L}$ is the sum of the loss values on all branches.

During test, for the branch $\mathbf{B^i}$, after we get the output $Z^{\mathbf{i}}$, we can get the **indicator vector** $\hat{\mathbf{Y}}$:

$$\hat{\mathbf{Y}} \in \mathbb{R}^N; \quad \hat{\mathbf{Y}}_{\mathbf{i}} = \underset{\mathbf{k}}{\operatorname{argmax}} \left\{ Z_{\mathbf{k}}^{\mathbf{i}} \right\} - 1; \, (\mathbf{k} \in \{1, 2\}) \tag{12}$$

The branch $\mathbf{B^i}$ finally produce a number of 0 or 1 based on $Z^{\mathbf{i}}$. When we get the number of 1 in the branch $\mathbf{B^i}$ that indicates the *presence* of the corresponding label $\mathbf{i}$, we say the model predicted that *there are instances with the label $i$ in the bag*. Hence, the indicator vector $\hat{\mathbf{Y}}$ describes whether the bag contains different labels. And at last,

$$\widetilde{\mathbf{Y}} = \underset{\mathbf{k}}{\operatorname{argmax}} \left\{ \hat{\mathbf{Y}}_{\mathbf{k}} \right\}$$
$$\text{s.t. } \hat{\mathbf{Y}}_{\mathbf{k}} = 1; \, \mathbf{k} \in [1, N] \tag{13}$$

When different branches output 1, we take the highest branch, because we need to know the maximum label in the bag as shown in Equation 13. Above equations produce the final result $\widetilde{\mathbf{Y}}$, $\widetilde{\mathbf{Y}} \in [\mathbf{0}, \mathbf{N}]$.

## 4 Experiments and Analyses

### 4.1 Bag-Level Performance: Evaluation on Benchmarks

With the branch number $N = 1$, five classic MIL benchmark datasets [3][1] are used to evaluate the bag-level performance, and each provides the samples in form of bags, where the bag size $t$ varies widely, ranging from 1 to 1044. Experimental results are shown in Table 1. It is a comprehensive test for the model under different conditions. From Table 1, it is obviously seen that our method (RGMIL) obtains the highest accuracies for five datasets, especially for FOX dataset. More benchmark performances are shown in Section B of our supplementary.

| Methods | MUSK1 | MUSK2 | FOX | TIGER | ELEPHANT |
|---|---|---|---|---|---|
| MI-Net(2018) | $0.887 \pm 0.041$ | $0.859 \pm 0.046$ | $0.622 \pm 0.038$ | $0.830 \pm 0.032$ | $0.862 \pm 0.034$ |
| ABMIL(2018) | $0.892 \pm 0.040$ | $0.858 \pm 0.048$ | $0.615 \pm 0.043$ | $0.839 \pm 0.022$ | $0.868 \pm 0.022$ |
| Gated-ABMIL(2018) | $0.900 \pm 0.050$ | $0.863 \pm 0.042$ | $0.603 \pm 0.029$ | $0.845 \pm 0.018$ | $0.857 \pm 0.027$ |
| DP-MINN(2018) | $0.907 \pm 0.036$ | $0.926 \pm 0.043$ | $0.655 \pm 0.052$ | $0.897 \pm 0.028$ | $0.894 \pm 0.030$ |
| Non-Local(2018) | $0.921 \pm 0.017$ | $0.910 \pm 0.009$ | $0.703 \pm 0.035$ | $0.857 \pm 0.013$ | $0.876 \pm 0.011$ |
| Asymmetric Non-Local(2019) | $0.912 \pm 0.009$ | $0.822 \pm 0.084$ | $0.643 \pm 0.012$ | $0.733 \pm 0.068$ | $0.883 \pm 0.014$ |
| Dual-Stream MIL(2021)* | $0.932 \pm 0.023$ | $0.930 \pm 0.020$ | $0.729 \pm 0.018$ | $0.869 \pm 0.008$ | $0.925 \pm 0.007$ |
| BDR(2022) | $0.926 \pm 0.079$ | $0.905 \pm 0.092$ | $0.629 \pm 0.110$ | $0.869 \pm 0.066$ | $0.908 \pm 0.054$ |
| GAMIL(2023) | $0.933 \pm 0.065$ | $0.910 \pm 0.085$ | $0.685 \pm 0.093$ | $0.894 \pm 0.070$ | $0.915 \pm 0.058$ |
| RGMIL | $\mathbf{0.968 \pm 0.060}$ | $\mathbf{0.985 \pm 0.039}$ | $\mathbf{0.954 \pm 0.048}$ | $\mathbf{0.951 \pm 0.045}$ | $\mathbf{0.965 \pm 0.032}$ |

Table 1: The classification accuracy (mean ± std) of RGMIL and compared methods on five benchmark datasets (*for S-O-T-A method). The results are the average of five independent experiments with a 10-fold cross-validation. Statistics collected from MI-Net[27], ABMIL[12], Gated-ABMIL [12], DP-MINN[29], Non-Local[26], Asymmetric Non-Local[34], DSMIL[15], BDR[11] and GAMIL[32].

## 4.2 A Challenging Dataset for Demonstration

Since pain estimation is generally regarded as the classification problems with four classes, we need to set the branch $N$ to 3. It indicates that we construct a 3-branch RGMIL to meet the upcoming challenges. Due to the extremely imbalance of label distribution, the pain data is not really a clear and flexible enough choice to present a demonstration. Thus, we constructed a flexible MIL dataset based on MNIST [14] and denoted it as MMNIST. In MMNIST, samples are presented in form of bags. A bag is composed of $\mathbf{t}$ grayscale images (1,28,28) selected from the MNIST dataset. With $\mathbf{N} = \mathbf{3}$, we take the images with label 0 to 3 to form MMNIST bags. The bag label is assigned based on the label with the largest count. The number of training bags for each bag label is approximately 1000. The labels in the bag are uniformly distributed. For the test, we use a fixed number of 10000 test *images*.

## 4.3 Evaluation Methods on the Aggregator $\rho$

We introduce the *error rate $\phi$*, and the *distance $\gamma$* to describe the aggregator $\rho$ to figure out how good the aggregator $\rho$ describes a MIL problem.

To explain the indicators, we still need to start from a classical MIL binary-classification scenario. Given the instance representations $\mathbf{fs} = [\mathbf{f}_1, \mathbf{f}_2, \ldots, \mathbf{f}_t] \in \mathbb{R}^{c \times t}$, the aggregator $\rho$, the classifier with linear transformation $\mathbf{A}$ and the sigmoid function $\text{sigm}(\ldots)$ as the activate function, with the total $\mathbf{N}$ bags with the bag label $\mathbf{Y} \in [0, 1]$ in the dataset. Suppose it be currently in the training phase, the training phase is equivalent to a process that we try to find the solution to the system of $\mathbf{N}$ equations, each with the following formula:

$$\text{sigm}(\mathbf{A}\rho(\mathbf{fs})) = \mathbf{Y}; \mathbf{Y} \in \{0, 1\} \tag{14}$$

Here, we assume that aggregator $\rho$ can be represented in a special formula:

$$\rho(\mathbf{fs}) = \sum_{k=1}^{t} \alpha_k \mathbf{f}_k; \text{ s.t. } \sum_{k=1}^{t} \alpha_k = 1; \alpha_k \geq 0 \tag{15}$$

Then, we can transform the Equation 14 into the following formula:

$$Z = \sum_{k=1}^{t} \alpha_k \cdot (\mathbf{A}\mathbf{f}_k) = \begin{cases} +\infty; & \text{if the given bag label is 1} \\ -\infty; & \text{otherwise} \end{cases} \tag{16}$$

The detailed explanation of this transform are given in Section A of our supplementary. Then, we consider the training process when we apply the gradient-descent algorithm here. Suppose the corresponding weight $\alpha_k$ of the $k$th instance in a positive bag be larger than others', the absolute value of $\frac{\partial Z}{\partial (\mathbf{A}\mathbf{f}_k)}$ *would be larger than others'*. Here we think the instance produces a large gradient. We recognize that $\mathbf{A}\mathbf{f}_k$ is the classification score of the corresponding $k$th instance. What's more, if its instance label is consistent with the bag label, we call it a *critical instance*. Given $\alpha_k$ greater than 0, the gradient-descent algorithm would just push the parameters $\mathbf{A}\mathbf{f}_k$ to *the right direction to produce a correct instance-level prediction*. The overall gradient of the parameters in the network will be the sum of different gradients produced by different instances in this bag. In conclusion, if an instance in a bag has *a large corresponding* weight $\alpha$ and it is *a critical instance*, then Equation 16 would be

| Aggregators | **10**/1 | 16/1 | 32/1 | 48/1 | **64 / 1** | **512 / 1** | 1/1(**Full**) |
|---|---|---|---|---|---|---|---|
| MXP | 93.47 | 88.71 | 81.17 | 73.02 | 68.11 | 65.37 | |
| ABP | 92.18 | 89.69 | 90.84 | 86.59 | 89.47 | 42.21 | |
| G-ABP | 91.47 | 89.13 | 87.47 | 84.38 | 86.93 | 36.99 | **97.75** |
| DSP | 58.54 | 58.12 | 58.47 | 56.13 | 52.43 | 42.23 | |
| RGP | **98.25** | **97.55** | **96.37** | **95.12** | **94.50** | **87.28** | |

Table 2: Instance-level Average Test Accuracies for the first 30 Training Epochs on MMNIST. The first row contains the different modes (M/N), and 1/1(Full) expresses the fully supervised case.

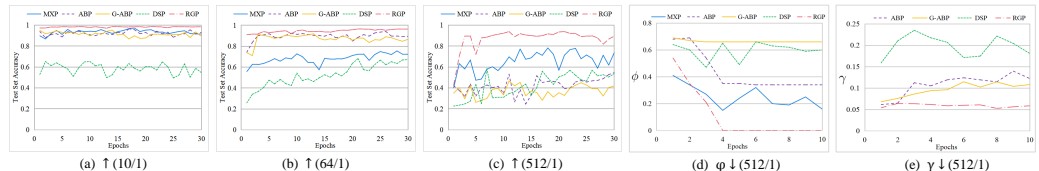

(a) ↑(10/1)   (b) ↑(64/1)   (c) ↑(512/1)   (d) $\varphi$↓(512/1)   (e) $\gamma$↓(512/1)

Figure 2: Detailed metric fluctuations of Table 2 under different modes (M/N).

likely to produce a right gradient-descent direction to achieve good performance on instance-level prediction, and that is what we want. If the gradient-descent direction of the equation is beneficial for the instance-level prediction, we call it a *right equation*, otherwise an *erroneous equation*. The *error rate* $\phi$ is the percentage of erroneous equations to the total number $N$ of equations. $\phi$ describes *the degree of deviation during the training phase*. When $\phi > 0$, gradient-descent accumulates errors in the wrong direction. In practice, we approximate $\phi$ by following steps: In a positive bag, we calculate the values of $\alpha$ for critical instances and take the average as $\beta1$, and the average value of $\alpha$ for non-critical instances is denoted as $\beta2$. If $\beta1$ is smaller than $\beta2$, we consider this to be an *erroneous equation*. We approximate $\phi$ by taking the ratio of the number of erroneous equations and the total number of positive bags, and $\phi$ is evaluated once for every training epoch. For a fully supervised model, it is easy to imagine that its $\phi$ is 0 from the beginning of train phase, since every equation would produce a right gradient-descent direction. We argue that *with a good aggregator $\rho$, $\phi$ is supposed to be close enough to 0, to be similar enough to a fully supervised model*.

Another indicator is the *distance $\gamma$*. Still imagine what human will do when facing the MIL problem. To predict whether the bag is positive, we usually make decisions *only based on the several critical instances* in this bag. A well-designed $\rho$ should simulate this characteristic. $\gamma$ is a metric that measures the difference in weights between critical instances in a positive bag, which describes how similar it is to human decision-making. In practice,we calculate $\gamma$ as follows: For every *positive bag*, we calculate the difference between the maximum and minimum weights $\alpha$ of the corresponding *critical instances*, and take the average of these differences for all positive bags, denoted as $\gamma$. $\gamma$ is also evaluated once for every training epoch. For the max pooling, it is easy to imagine that its distance $\gamma$ could reach 1, but we do not want the distance to be large. We argue that *with a good aggregator $\rho$, $\gamma$ is supposed to be close enough to 0, to be similar enough to how we make decisions*.

Notice that $\phi$ and $\gamma$ are introduced to *describe the training process* for demonstration, thus all *evaluated on the training set*. **Instance labels are known just when we evaluate $\phi$ and $\gamma$**.

### 4.4 Instance-Level Evaluation on MMNIST: Why RGP Works?

When an experiment with the training bag size (**M**) and the testing bag size (**N**) is implemented, we define it as the **M/N** mode. We give comprehensive experiments in different modes to test the aggregators in Table 2. For some detail fluctuations see Fig. 2. Results show that RGP shows dominance in different scenarios. As shown in Fig. 2 (d) and (e) under mode 512/1, *with the faster convergence of $\phi$ to 0 and the closer value of $\gamma$ to 0, RGP exhibits a working logic that is sufficiently similar to human thinking process and a supervised algorithm*. For RGP, it represents the consistency between instances and bags through specific structure, so that the learning process can be easily transferred to instance level. Attention-based thoughts are valuable, but simply applying more complicated attention modules to analyze the problem, as done in ABP series, may not lead to breakthrough results on instance level because indicators show that they struggle to catch the points. Another critical fact we found is that if we switch training bag size to 1, which would bring a fully

| Branch | 0 | 1 | 3 | 2 | 0 | 2 | 2 | 1 | 1 | 3 | $\hat{\mathbf{Y}}$ |
|---|---|---|---|---|---|---|---|---|---|---|---|
| 3 | 0.0151 | 0.0351 | **0.4981** | 0.1183 | 0.0220 | 0.0357 | 0.0365 | 0.0601 | 0.0313 | **0.1497** | 1 |
| 2 | 0.0132 | 0.0577 | 0.0841 | **0.3062** | 0.0185 | **0.1959** | **0.1977** | 0.0299 | 0.069 | 0.0360 | 1 |
| 1 | 0.0375 | **0.1889** | 0.0879 | 0.0479 | 0.0401 | 0.0793 | 0.0889 | **0.2001** | **0.1732** | 0.0563 | 1 |

Table 3: RGP Information Table (64/10) on MMNIST. The first row contains the 10 images of the test bag. Indicator vector $\hat{\mathbf{Y}}$ is the model output. The remaining contents are all the weights of corresponding instances from all the branches.

supervised scenario, the performance is lower than RGP in 10/1 mode. This clearly tells that **RGP is not a performance bottleneck, it would enhance the model when the bag size is relatively small**. With the guidance of regressor, RGP describes the MIL problem accurately, which make it similar enough to a supervised case to produce discriminative instance-level representations from backbone. We can also say that, *when the number of data records(can be in form of bags or instances) is given fixed, MIL model with RGP could even outperform a fully supervised one under some condition.* That verifies our **Assumption 1**.

We also provide a special experiment under mode 64/10, with the RGP information table presented in Table 3. We see that not only we get a correct indicator vector $\hat{\mathbf{Y}}$ on bag level, we can also directly provide instance-level predictions by combining weight-related information with the model output $\hat{\mathbf{Y}}$: try to take the maximum value in each column.

RGP strictly follows the expectation we gave it, which is a strong constraint inspired by the inference rule. For a positive bag, according to the way RGP works, when we are in the training process, the $k$th instance representation with a high classification score inside the bag **MUST** have a larger value of weight $\alpha_k$ . Referring to Equation 16 that explains $\phi$ and $\gamma$, the absolute value of its partial derivative $\frac{\partial Z}{\partial (\mathbf{A}\mathbf{f}_k)}$ during the training process would be larger. The gradient descent algorithm will increase the classification score, which promotes the increase of its weight $\alpha_k$ because it explicitly depends on the classification score in RGP. The gradient descent process produces instance-level representations with *large weights and high classification scores* in positive bags. For negative bags, the training process will lower the classification scores of **all** instance representations inside it, but since there are no positive instances in it, the weights of the instance representations would be random. The gradient descent process will uniformly decrease the classification scores of all instances inside the negative bag. **In the end of training process, RGMIL will converge to a solution where all the training bags are predicted correctly. In this final state, there is a huge difference in instance weights, and instance representations in positive bags with high classification scores will have significantly larger weights.** This final solution we get from RGMIL would comply with our inference rule above. **In a MIL problem, any solution that meets our inference rule is legal.**

Since we have verified that the final convergence state of RGMIL is a legal solution to the MIL problem, the convergence process implies an improvement in the model's ability to distinguish critical instances. But still, the speed of convergence is crucial to improve instance-level performance. We must accelerate the speed at which the model generates discriminative instance-level representations as much as possible, because when the loss value reaches or approaches 0, the model wouldn't be trained anymore, and too slow a speed can stop the model from reaching the final solution. With low $\gamma$ value, RGP is able to capture more critical instances, which brings a quicker improvement in instance-level discriminative ability. Based on this, RGMIL could correct previous erroneous inferences to reduce the $\phi$ value quickly and thus avoid the accumulation of errors on instance level. That leads to a huge improvement on instance-level performance when compared with other methods. After three epochs of training, RGMIL successfully lowered the $\phi$-value to 0 in the fourth epoch and maintained it till the end. Multiple experiments have shown that RGMIL can always adjust its inference to lower the $\phi$-value to 0 within the first seven training epochs in 512/1 mode.

### 4.5 Ablation Study of RGP

As shown above, we have provided the derivation of the feasibility of RGP using sigmoid as the activation function, which we refer to as the original version. The derivation is built on the case where there is no trouble in numerical computation or gradient descent. However, in the practical implementation, we still need to explore a more practical way. Here, we still refer to the formula of the RGP forward propagation process in the $i$th branch ($\mathbf{B^i}$) mentioned in previous section. In

this implement, we take the difference by using Equation 6, meanwhile using Equation 7 to achieve the normalization for preventing gradient vanishing. Without the normalization equation, if we take the value of the second bit of the softmax output instead of the difference, it would be similar to the original version, which we refer to as the baseline version. The experimental results for different versions are shown in Table 4. Obviously, it is seen that the performance based the different and the normalization is superior to others. Without the training tricks from the equations, it is difficult to reduce $\gamma$-value to a low level.

| Aggregators | $10/1$ | $32/1$ | $64/1$ | $512/1$ |
|---|---|---|---|---|
| RGP (B) | 94.60 | 82.64 | 62.75 | 66.54 |
| RGP (B + N) | 95.31 | 93.92 | 90.88 | 72.70 |
| RGP (B + N + D) | **98.25** | **96.37** | **94.50** | **87.28** |

Table 4: Instance-level Average Test Accuracies for first 30 Training Epochs on MMNIST, where 'B' expresses Baseline Version, 'N' expresses Normalization Formula, 'D' expresses Difference Formula.

## 4.6 Comparison with Supervised Models in Pain Estimation

In this part, pain estimation is considered as a real application to validate the performance of RGMIL, where the benchmark pain dataset (UNBC-McMaster Shoulder Pain dataset [17], UNBC) is used. The UNBC dataset contains 200 video sequences with 48398 frames from 25 subjects, and all frames are well-annotated. To compare with the *existing supervised models*, we follow the same experimental settings with the S-O-T-A model MSRAN (ICBBT'21) [2]: using 25-fold cross-validation with a leave-one-subject-out strategy and evaluating by four metrics. In experiment, all images are resized as (3, 224, 224), and the training bag size is set as 64. An average of about 6000 bags is used as training data during each fold validation, and ResNet18 [8] is used as backbone. Experimental results are shown in Table 5. From Table 5, it is seen that RGMIL does gain *comparable* performance to supervised methods. It indicates that *current design is effective enough to be practical without causing any performance degradation compared to supervised ones for pain estimation.* More details about metrics and settings are given in Section C of supplementary.

| Methods | MAE $\Downarrow$ | MSE $\Downarrow$ | PCC $\Uparrow$ | ICC $\Uparrow$ |
|---|---|---|---|---|
| Deep Pain(2017) | 0.50 | 0.74 | 0.78 | 0.45 |
| DSHF(2018) | - | 0.94 | 0.68 | - |
| DBR(2018) | - | 0.69 | **0.81** | - |
| Multistream CNN(2019) | 0.47 | 0.53 | 0.70 | 0.55 |
| MSRAN*(2021) | 0.40 | 0.46 | 0.78 | **0.63** |
| LIAN(2021) | 0.45 | 0.66 | 0.81 | 0.61 |
| RGMIL | **0.31** | **0.42** | 0.77 | 0.62 |

Table 5: Instance-level Performance Comparison with Supervised Models (*:S-O-T-A model) on UNBC pain dataset, where all results are the mean of 25-fold cross-validation and four metrics are used: MAE, MSE, PCC and ICC. In this experiment, we take consecutive 64 frames in a video as a training bag. To increase the amount of data, we used a sliding window approach with a step size of 8 to produce training bags. Statistics collected from Deep Pain [21], DSHF [25], DBR [23], Multistream CNN [10], MSRAN [2], LIAN [28].

## 4.7 Discussion for More General Scenarios and Limitations

To validate the performance of RGMIL on the general multi-class bag-level classification problems, we test our model on SIVAL [20] dataset, where SIVAL consists of 25 classes of complex objects photographed in different environments and each class contains 60 images. In this experiment, each image is segmented into approximately 30 segments, and each segment is represented by a 30-dimensional feature vector that encodes information. The segments are labeled as containing the object or the background. We separately select 1, 3 and 10 classes as positive classes, and randomly sample from the other classes as the negative class. We set up three Linear+ReLU blocks as feature extractors. The test scenario involves image classification problems ranging from binary to 11-class classification. Here, each image is treated as a bag. Considering that the classes of SIVAL do not have any ordering relationship, all bits of the output indicator vector are reliable, thus all branches are involved to the computation of loss function. Continuing with the current version's approach, the

model's output during test is still obtained by selecting the branch with the highest position among all branches with an output of 1. Table 6 shows the bag-level performance on SIVAL dataset [20]. From Table 6, it is seen that RGP still obtains competitive results on the general bag-level problem.

| # of positive classes | 1 | 3 | 10 |
|---|---|---|---|
| MXP | 94.87 | 89.08 | 60.95 |
| ABP | 97.05 | 93.93 | 75.71 |
| G-ABP | **97.31** | 94.28 | **81.33** |
| DSP | 96.79 | 89.60 | 75.91 |
| RGP | **97.31** | **94.40** | 80.19 |

Table 6: Bag-level Results on the general multi-class image classification problem (the SIVAL dataset), where 1, 3 and 10 express three different multi-classification problems in which 1, 3 and 10 classes are separately used as positive classes, respectively. The shown result is the average of 10-time test accuracy for each model after training convergence.

Moreover, we also validate RGMIL on the general multi-class instance-level classification problems. In this experiment, we still use the 3-branch RGMIL, and adopt a different approach in constructing the data: the images from classes (ranging from 0 to 6) in MMNIST are considered as the negative instances, like the background class, and the images from classes (7, 8 and 9) in MMNIST are as three different positive instances. Table 7 shows the instance-level performance of the constructed four classes classification on MMNIST. Unlike the original MMNIST series, the convergence speed of different methods varies much. Thus, we present the average of 10-time test accuracy for each model after convergence or training after 100 epochs. Each positive class and the negative class account for approximately 25% of the data. Within one positive bag, the number of images of positive class is approximately 10%. The processing of loss value and output is the same as the SIVAL experiment above. From Table 7, it is also seen that instance-level results are promising in a more general case.

| Aggregators | 10/1 | 16/1 | 32/1 | 64/1 | 256/1 | 512/1 |
|---|---|---|---|---|---|---|
| MXP | 76.09 | 71.78 | 57.49 | 58.15 | 38.63 | 32.48 |
| ABP | 71.76 | 75.17 | 71.81 | 68.33 | 69.09 | 62.43 |
| G-ABP | 73.12 | 75.04 | 72.21 | 64.58 | 69.20 | 60.56 |
| DSP | 62.81 | 69.29 | 63.75 | 67.04 | 59.77 | 52.91 |
| RGP | **88.79** | **85.73** | **85.02** | **83.19** | **82.50** | **81.46** |

Table 7: Instance-level Results on general 4-class problems constructed based MMNIST. The first row contains the different modes (M/N). The shown result is the average of 10-time test accuracy for each model after convergence or 100 epochs.

Although RGMIL is suitable to solve the general multi-classification problem in which class dependencies or ordering need not be considered, the feasibility of training on instance-level with an excessively large bag-length still need to be ensured. As shown in Table 1, it is seen that the instance-performance decays with increasing bag lengths. Obviously, when the bag length is 512, the accuracy is lowest and reduced by 11% than the minimum bag length. In addition, there is memory limitation to handle the excessively large bags when the feature extractor is involved in training.

## 5 Conclusion

In this paper, we propose a new vision on MIL with a practical multi-classification MIL model (RGMIL). RGMIL extends the classical binary to a multi-classification scenario via *optimizing several reliable equations simultaneously*. More importantly, a key component (RGP) is introduced in the MIL model. We argue that the difference between a supervised problem and MIL is **deterministic and clear**: we may not need an over-complicated black box to simulate the difference. An accurate description exists. We describe the characteristic of MIL problems through the structure of RGP, and *only the parameters in the feature extractor (backbone) need to be trained* to provide the discriminative instance-level representation. This is a method that has good interpretability and conforms to the true inference rule of the MIL problem. We also validate RGMIL in real-world application on the UNBC dataset. *Suffice it to say that it is possible for a MIL model to achieve even better instance-level performance than some supervised models.* We expect RGP to become an important component of future MIL network architectures and to provide some inspiration.

## Acknowledgments

This work is supported in part by the State Key Program of National Natural Science of China under Grant 62234010, the National Natural Science Foundation of China under Grant 61806154, the China Postdoctoral Science Foundation Funded Project under Grant 2019M653565, and the Fundamental Research Funds for the Central Universities No.ZYTS23060.

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
