## A. About Equation 16

Now we are at training phase, which is a process that we try to find the solution to the system of $\mathbf{N}$ equations, each like this:

$$\mathrm{sigm}(\mathbf{A}\rho(\mathbf{fs})) = \mathbf{Y}; \mathbf{Y} \in \{0, 1\} \tag{1}$$

Here we assume that aggregator $\rho$ can be represented in such a special form:

$$\rho(\mathbf{fs}) = \sum_{k=1}^{t} \alpha_k \mathbf{f}_k; \text{ s.t. } \sum_{k=1}^{t} \alpha_k = 1; \alpha_k \geq 0 \tag{2}$$

It is practical to find the expression of $\alpha_k$ in some typical cases like *max pooling, average pooling, attention-based pooling and its variations*, while in a general case it may be tough to find the accurate expression. *We only consider the feasible cases*. Then we consider the $\mathrm{sigm}(\ldots)$ function in Equation 1. For instance, say x is the parameter to be trained, we want x to satisfy $\mathrm{sigm}(x) = 1$ or 0, we would apply gradient-descent which will cause $x \to +\infty / -\infty$. And also notice that both $\mathbf{A}$ and $\rho$ are linear operators, so we could then transform the Equation 1 into the following equivalent form:

$$Z = \sum_{k=1}^{t} \alpha_k \cdot (\mathbf{A}\mathbf{f}_k) = \begin{cases} +\infty; & \text{if given bag label is 1} \\ -\infty; & \text{otherwise} \end{cases} \tag{3}$$

## B. Benchmark Performances on Bag Level

Two basic blocks of fully connected + ReLU layer are utilized as *backbone* for all benchmark datasets, generating instance-level representations with a dimensionality of 512. We set one branch here for the binary-classification problems. The benchmark experiments are all performed on bag level.

We adopt the same experimental settings from the original DSMIL paper codes[5], and provide DSMIL performances as comparisons. Experiments were run 5 times each with a 10-fold cross-validation on all benchmark datasets. For each fold we train the model for 40 epochs then evaluate the performance. We report the mean and standard deviation of the classification accuracy (mean ± std) of DSMIL and RGMIL.

Major difference here between RGMIL and DSMIL: **RGMIL would not train any parameters other than the *instance-level backbone*, while DSMIL takes fixed features from datasets and only trains parameters in the aggregation and regression stage.**

Detailed properties of 5 benchmark datasets used in paper:

| DATASET | MUSK1 | MUSK2 | Elephant | Fox | Tiger |
|---|---|---|---|---|---|
| dimension | 166 | 166 | 230 | 230 | 230 |
| # of bags | 92 | 102 | 200 | 200 | 200 |
| # of positive bags | 47 | 39 | 100 | 100 | 100 |
| # of instances | 476 | 6598 | 1391 | 1320 | 1220 |
| max bag size | 40 | 1044 | 13 | 13 | 13 |
| min bag size | 2 | 1 | 2 | 2 | 1 |

### Webpages

The problem is to classify webpage as interesting or not[6]. In total, 9 users rate webpages as such, therefore there are 9 different datasets. A webpage is a bag, and the links on the webpage are the instances. The features are related to word frequency (and therefore very high-dimensional). There are 113 bags in each dataset, with bag size varying from 4 to 200. Each dataset contains 3423 instances.

| DATASET | instance dimension | DSMIL | RGMIL |
|---|---|---|---|
| Web1 | 5863 | $0.847 \pm 0.190$ | $1.0 \pm 0.0$ |
| Web2 | 6519 | $0.856 \pm 0.169$ | $1.0 \pm 0.0$ |
| Web3 | 6306 | $0.875 \pm 0.135$ | $1.0 \pm 0.0$ |
| Web4 | 6059 | $0.781 \pm 0.155$ | $1.0 \pm 0.0$ |
| Web5 | 6407 | $0.768 \pm 0.182$ | $1.0 \pm 0.0$ |
| Web6 | 6417 | $0.784 \pm 0.130$ | $1.0 \pm 0.0$ |
| Web7 | 6450 | $1.0 \pm 0.0$ | $1.0 \pm 0.0$ |
| Web8 | 5999 | $1.0 \pm 0.0$ | $1.0 \pm 0.0$ |
| Web9 | 6279 | $0.994 \pm 0.022$ | $1.0 \pm 0.0$ |

## 20NewsGroups

20NewsGroups[7] contains a collection of 20 datasets where each of them is consisted of 100 bags. Each bag contains approximately 40 instances of articles from 20 different topics where each instance represents one article described by the top 200 TF-IDF features. There are 50 positive bags in each dataset. A bag is considered to be positive if at least one of its instances belongs to a specific topic. However, in each positive bag only approximately 3% of the instances are positive.

| DATASET | # of instances | DSMIL | RGMIL |
|---|---|---|---|
| alt.atheism | 5443 | $0.700 \pm 0.458$ | $1.0 \pm 0.0$ |
| comp.graphics | 3094 | $0.740 \pm 0.439$ | $1.0 \pm 0.0$ |
| comp.os.ms | 5175 | $0.720 \pm 0.449$ | $1.0 \pm 0.0$ |
| comp.sys.ibm | 4827 | $0.720 \pm 0.449$ | $1.0 \pm 0.0$ |
| comp.sys.mac | 4473 | $0.700 \pm 0.458$ | $1.0 \pm 0.0$ |
| comp.window.x | 3110 | $0.778 \pm 0.413$ | $1.0 \pm 0.0$ |
| misc.forsale | 5306 | $0.716 \pm 0.447$ | $1.0 \pm 0.0$ |
| rec.autos | 3458 | $0.800 \pm 0.400$ | $1.0 \pm 0.0$ |
| rec.motorcycles | 4730 | $0.704 \pm 0.445$ | $0.970 \pm 0.090$ |
| rec.sport.baseball | 3358 | $0.720 \pm 0.449$ | $1.0 \pm 0.0$ |
| rec.sport.hockey | 1982 | $0.770 \pm 0.415$ | $1.0 \pm 0.0$ |
| sci.crypt | 4284 | $0.802 \pm 0.396$ | $0.994 \pm 0.024$ |
| sci.electronics | 3192 | $0.858 \pm 0.346$ | $0.996 \pm 0.020$ |
| sci.med | 3045 | $0.700 \pm 0.458$ | $1.0 \pm 0.0$ |
| sci.religion | 4677 | $0.700 \pm 0.458$ | $1.0 \pm 0.0$ |
| sci.space | 3655 | $0.728 \pm 0.435$ | $1.0 \pm 0.0$ |
| talk.politics.guns | 3588 | $0.640 \pm 0.480$ | $1.0 \pm 0.0$ |
| talk.politics.mideast | 3376 | $0.760 \pm 0.427$ | $0.998 \pm 0.014$ |
| talk.politics.misc | 4788 | $0.692 \pm 0.454$ | $1.0 \pm 0.0$ |
| talk.religion.misc | 4606 | $0.702 \pm 0.455$ | $1.0 \pm 0.0$ |

## Messidor

Messidor[1, 3] is an image classification problem. The data consists of 1200 eye fundus images from 654 diabetes and 546 healthy patients. Each image from the original data is rescaled to 700×700 pixels and split up into patches of 135×135 pixels. Patches which do not have a sufficient amount of foreground are discarded. The features used are: intensity histogram of RGB channels for 26 bins, mean of local binary pattern histograms of 20×20 pixel grids, mean of SIFT descriptors, and box count for grid sizes 2 to 8. Some of the features return NaNs, replacing by zero is advised. Accuracy±standard deviation of DSMIL is $0.994 \pm 0.009$, and for RGMIL it's $0.998 \pm 0.006$ .

| DATASET | # of bags | # of instances | dimension | max/min bag size |
|---|---|---|---|---|
| Messidor | 1,200 | 12,352 | 687 | 12/8 |

## UCSB Breast

UCSB Breast[4] is an image classification problem. The original datasets consists of 58 TMA image excerpts (896 × 768 pixels) taken from 32 benign and 26 malignant breast cancer patients. The

learning task is to classify images as benign (negative) or malignant (positive).Patches of 7×7 size are extracted. The image is thresholded to segment the content from the white background and the patches that contain background more than 75% of their area are discarded. The features used are 657 handcrafted features that are global to the patch, and averaged features extracted from the cells, detected in each patch. Accuracy±standard deviation of DSMIL is $0.915 \pm 0.118$, and for RGMIL it's $0.955 \pm 0.077$.

| DATASET | # of bags | # of instances | dimension | max/min bag size |
|---|---|---|---|---|
| UCSB Breast | 58 | 2,002 | 708 | 40/21 |

## C.   Detail on UNBC Experiments

Resnet18[2] without the last full-connection layer is used as backbone, which would produce a 512-dimension representation for each UNBC image. We configure branch number **N** to be 3 for the 4-class classification scenario. Use the Adam optimizer to perform the gradient descent algorithm with a fixed learning rate of 0.0005. Experiments were run with a 25-fold cross-validation. For each fold we train the model for 40 epochs then evaluate the performance. The four evaluation metrics are as follows:

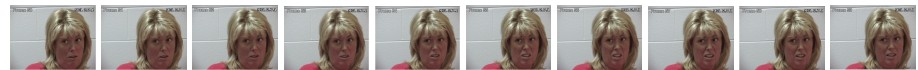

Figure 1: UNBC raw video clip. Image shape:(3,320,240)

MAE (Mean Absolute Error), which can be seen as L1 loss, is a commonly used loss function for regression models. It is the sum of the absolute differences between the target values and the predicted values.

MSE (Mean Squared Error), which can be seen as L2 loss, is also a widely used regression loss function. It is the sum of the squared differences between the predicted values and the true values. By squaring the errors (letting = true value - predicted value), MSE amplifies the errors if > 1. If there are outliers in the data, the value of can be large, and thus squared will be much larger than | |. Therefore, compared to using MAE to compute the loss, using MSE gives more weight to the outliers.

We use these two metrics to measure the overall error between the predicted values and the ground truth at the frame level during test. Lower values of MAE and MSE indicate more accurate predictions by the model. In the UNBC experiments, due to the imbalanced distribution of pain levels, a large number of frames have a pain level of 0. Therefore, if accuracy metric is used for model evaluation, it can easily overestimate the model performance. Hence, accuracy metric is generally not used for model evaluation.

PCC(Pearson Correlation Coefficient) measures the linear correlation between two variables, typically the predicted values and the ground truth. It ranges from -1 to 1, where a value close to 1 indicates a strong positive linear correlation, a value close to -1 indicates a strong negative linear correlation, and a value close to 0 indicates no linear correlation. PCC is widely used to assess the overall relationship or agreement between two continuous variables.

ICC (Intraclass Correlation Coefficient) is a statistical measure that quantifies the consistency or reliability of measurements made by different observers or raters. It is commonly used when there are multiple raters or multiple measurements taken on the same subjects. ICC ranges from 0 to 1, where a value close to 1 indicates high agreement or consistency among the raters or measurements, and a value close to 0 indicates low agreement.

MAE (Mean Absolute Error) and MSE (Mean Squared Error) are loss functions that quantify the error or discrepancy between predicted and true values. They are used during model training and optimization. Lower values of MAE and MSE indicate better model performance. PCC and ICC, on the other hand, are evaluation metrics that assess the agreement, correlation, or consistency between predicted and true values. They provide insights into the quality of predictions and the reliability of measurements. Higher values of PCC and ICC indicate better agreement or consistency.

Most of the supervised models use some of the above metrics to evaluate performance on pain estimation problems. With the sampe experimental settings, our model demonstrates similar performance to fully supervised models on these metrics, indicating comparable reliability in pain estimation tasks.

The detailed results are as follows:

| Test ID | PCC | ICC | MAE | MSE |
|---|---|---|---|---|
| 0 | 0.8359 | 0.7115 | 0.1854 | 0.1525 |
| 1 | 0.8343 | 0.7373 | 0.1218 | 0.0931 |
| 2 | 0.7554 | 0.6822 | 0.1302 | 0.0941 |
| 3 | 0.6734 | 0.3367 | 1.7363 | 1.1616 |
| 4 | 0.7368 | 0.3160 | 0.3158 | 0.2947 |
| 5 | 0.6712 | 0.6188 | 0.1021 | 0.0583 |
| 6 | 0.8196 | 0.6269 | 1.5688 | 1.1558 |
| 7 | 0.8967 | 0.8163 | 0.1827 | 0.1262 |
| 8 | 0.8278 | 0.5652 | 0.2266 | 0.2145 |
| 9 | 0.8074 | 0.5416 | 0.3484 | 0.3474 |
| 10 | 0.9028 | 0.6782 | 0.3496 | 0.3082 |
| 11 | 0.8954 | 0.7131 | 0.9355 | 0.6179 |
| 12 | 0.7018 | 0.6481 | 0.1548 | 0.0937 |
| 13 | 0.6248 | 0.5196 | 0.1820 | 0.1385 |
| 14 | 0.9862 | 0.8458 | 0.0839 | 0.0829 |
| 15 | 0.8368 | 0.6770 | 0.4405 | 0.2789 |
| 16 | 0.5037 | 0.3531 | 0.3170 | 0.2068 |
| 17 | 0.9031 | 0.8175 | 0.2448 | 0.1869 |
| 18 | 0.9198 | 0.8595 | 0.1539 | 0.1291 |
| 19 | 0.7952 | 0.7251 | 0.1257 | 0.1004 |
| 20 | 0.8439 | 0.7900 | 0.0718 | 0.0612 |
| 21 | 0.5061 | 0.4613 | 0.0717 | 0.0563 |
| 22 | 0.6644 | 0.4610 | 1.0913 | 0.9748 |
| 23 | 0.4820 | 0.3121 | 0.9636 | 0.4907 |
| 24 | 0.8247 | 0.6121 | 0.5106 | 0.3806 |
| AVERAGE | 0.76996 | 0.61704 | 0.42459 | 0.31224 |