# OpenReview forum: "RGMIL: Guide Your Multiple-Instance Learning Model with Regressor"
_NeurIPS.cc/2023/Conference — NeurIPS 2023 poster_

### Official Review · Reviewer_A2Ve · 2023-06-30

**Soundness:** 3 good
**Presentation:** 1 poor
**Contribution:** 2 fair
**Rating:** 3
**Confidence:** 4

**Summary:**

The paper presents a new approach for Multiple Instance Learning (MIL) called Regressor-Guided MIL (RGMIL). The contribution of RGMIL is a new aggregation approach Regressor-Guided Pooling (RGP). In this approach, the aggregation part of the MIL model is split into N branches (for N + 1 classes, assuming class 0 is a negative class), where each branch is responsible for classifying a single class. The model is applied to a range of binary MIL problems (MUSK1, MUSK2, FOX, TIGER, ELEPHANT, Webpages, 20NewsGroups, Messidor, and UCSB Breast), and two max-based multi-class problems: MMNIST (proposed in this work) and UNBC shoulder pain estimation. The model mostly outperforms existing methods, and analysis is conducted as to why RGP is an effective pooling strategy.

**Strengths:**

**Originality**
1. The RGB pooling is an original approach, particularly with the double pass through the regressor.
2. Section 4.3 is also a novel investigation into the bottlenecks of MIL model training and can help understand why certain models succeed/fail.

**Quality**
1. The evaluation covers a range of datasets and shows strong results, outperforming most other methods in most cases.
2. An ablation study is used to show how performance changes when training on different sizes bags but only classifying single instance bags.

**Significance**
1. Given the strong performance of RGMIL, it would appear useful in MIL settings that use max-based problems, i.e., traditional binary MIL or problems like pain estimation.

**Weaknesses:**

**Originality**
1. I believe some elements of the proposed method are already well known concepts, e.g., equation 4 appears to be one hot encoding, and equation 8 is a softmax. Simplifying these in the methodology and focusing on the novel parts of the method will help understanding.

**Quality**
1. Some elements of the approach are lacking justification (see questions below).
2. The MMNIST dataset is proposed but then only evaluated in an Instance-level setting. A strong evaluation could involve looking at the performance when the training and test bag sizes are the same (i.e., 10/10, 16/16 etc.) as well as the instance-level setting.

**Clarity**
1. I found the paper very hard to follow and it took a considerable effort to unpick what the proposed method was actually doing.
2. It is not clear quite how the method works, particularly with regards to training. In Section 3.2, I am puzzled by the statement *"Parameters in both the RGP and regressor will NOT be involved in gradient descent"*. Please see questions below.
2. The work mentions how CNNs work and how humans would approach a MIL problem as justification for their approach. This makes the narrative harder to follow and detracts from the design of the novel pooling mechanism. I think this comparison would be best placed in the appendix, with more space allocated to discussing the model architecture and explaining how it works.
3. Figure and Table captions often contain little information, e.g., Table 2 does not state which dataset the model is trained on nor what the different columns means, Figure 2 should have y-axis labels, etc.
4. Some of the mathematical notation should be revisited to aid understanding. For example, is it necessary to have $M$ as the total number of instances and then $t$ as the number of instances per bag? Equations 5 through 9 jump through different notations, e.g., $fs$ for representations, then $H$, then $F_i$.
5. I would suggest another thorough proof-read to remove grammatical errors that make the work hard to follow.

**Significance**
1. My understanding is that this approach is only applicable to multi-class problems that have some max-based ordering of classes, e.g., in pain estimation, if branch 1 and 2 are both positive, then the prediction is class 2. I don't know how this would work for general multi-class problems (see questions below), which limits its significance.
2. I disagree with the final paragraph of Section 2.2 (comparing t < c and t > c). Instance-level performance is important for interpretability in both cases - just because there are a greater number of instances per bag does not mean instance-level performance becomes less important.



**Questions:**

1. Why does each branch of the aggregator produce two outputs? I can see the two outputs are used to calculate a difference in Equation 6, but don't understand why a single output with a sigmoid activation and a threshold of 0.5 isn't used. This needs to be explained in more detail.
2. If my understanding is correct, the regressor is used to calculate some form of attention weights (output of equation 8) and also classify the re-weighted instance representations. Is that correct? If so, what is the relationship to other MIL attention methods? It seems somewhat similar to the Additive MIL approach [1], which is not cited.
3. Is each branch regressor assigned random weights that are then fixed during training of the feature extractor? If so, what is the motivation behind this?
4. How would this approach work on multi-class problems that don't have some form of max-based ranking, e.g., SIVAL [2] or 4-MNIST-Bags [3]? My confusion lies in what happens when two or more branches all have positive indicator vectors $\hat{Y}$. In the max-based datasets used in this work, the maximum branch is taken as the overall prediction (equation 13). However, I'm not sure how this would work for other types of multi-class MIL datasets that don't use max-based classes. This limits the significance and scalability of the work.

[1] Javed, Syed Ashar, et al. "Additive MIL: intrinsically interpretable multiple instance learning for pathology." Advances in Neural Information Processing Systems 35 (2022): 20689-20702.
[2] Rahmani, Rouhollah, et al. "Localized content based image retrieval." Proceedings of the 7th ACM SIGMM international workshop on Multimedia information retrieval. 2005.
[3] Early, Joseph, Christine Evers, and Sarvapali Ramchurn. "Model Agnostic Interpretability for Multiple Instance Learning." In International Conference on Learning Representations. 2022.

**Limitations:**

1. The work does not address the limitations of the novel method or suggest any areas for future work.
2. The main limitation of the work is that it is very hard to follow and difficult to understand how the novel method works. This is frustrating as the results appear promising.
3. One potential limitation, discussed in the questions above, is how this could be applied to multi-class datasets that do not follow some form of max-based assumption.

Due to the difficult in understanding the approach and the lack of clarity in the method, I am leaning towards rejection as I don't think the method is reproducible from what is presented. However, there is certainly promise in the approach and the results appear strong.

---

> ### Author Rebuttal · Authors · 2023-08-09
>
> Thanks a lot for your many valuable comments. It’s great that you appreciate our originality and investigation of the MIL model.
>
> Comment 1：Weaknesses-Originality
>
> [R] We will revise the writing thoroughly according to all comments.
>
> Comment 2: Weaknesses-Quality
>
> [R] We agree with the importance of bag-level performance. We have given additional general multi-classification experiments both on bag and instance level in global rebuttal(see PDF).
>
> Comment 3: Weaknesses- Clarity
>
> [R]: 1. We will revise our descriptions to make paper  more readable.
>
> 2.More responses can be found in question parts.
>
> 3.As a starting point and important inspiration for designing the pooling method, we mentioned CNN here. Details have been put into the supplementary.
>
> 4.We have modified relevant tables and figures.
>
> 5.We have fixed inconsistencies in the mathematical notations.
>
> 6. We will  correct grammatical errors as suggested.
>
> Comment 4: Weaknesses- Significance
>
> [R]: 1. We have achieved experiments on general multi-class datasets (SIVAL and the new MMNIST series), and results are shown in the global rebuttal.
>
> 2. We understand and agree with your thoughts about the instance-level importance. As the results on MMNIST show, the performance decays with the training bag length increases. In supplementary (section A.1), we argue that there is only one additional deterministic inference process when a human is faced with the MIL scenario. Any inference result that complies with the inference rule is legal. We assume that given the fixed total data records number, if the bag size is too large, there may be too many correct possibilities in a bag, which could cause instance-level performance degradation. This is an unsubstantiated idea and will be removed from the article.
>
> Comment 5: Question-1
>
> [R] The detailed responses are shown in the global rebuttal.
>
> Comment 6: Question-2
>
> [R] Yes. Between ABP and RGP, there are some important similarities and differences. They share the same view regarding the reliance on feature weights when obtaining bag representations. This view has been validated in many MIL models. This is a significant similarity.
>
> Between RGP and the ABP, the main difference lies in how to obtain the weights and the perspective on the MIL problem itself. The ABP series often train matrices (related to $Q$, $K$, $V$) to obtain weights, which is not very direct. In our perspective, instance-level performance is crucial, and we essentially adjust our bag-level judgment based on the current judgments of individual instances, as in human reasoning for MIL problem. In RGMIL, it believes that the solution to MIL problems should regress back to the instance level, and the criteria for judgment are the same on instance and bag levels, which is why we use the double-regressor scheme to pass the consistency. We believe that there is no need for a parameterized black-box model to simulate the reasoning process of a MIL problem. Simply asking for the current judgment from the instance-level regressor would be a more natural and intuitive way of aggregating features.
>
> RGP is non-parametric, and the learning process can entirely occur in instance feature extraction. Theoretical analysis in section 4.3 and section A.1 of supplementary confirms the advantage on instance level. In contrast, ABP does not take extra consideration on instance-level performance and the actual human reasoning process. This is the key difference between RGP and ABP.
>
> We have added the cite of Additive ABP [1]. For additive ABP, it only changes the order of summation and the output logits to improve interpretability. Essentially, it is considered as a member of the ABP series. Its motivation and research are quite different from our method.
>
> Comment 7: Question-3
>
> [R] Yes. In classification model, it is expected that obtained features are of linear separability. For example, in the architecture of supervised model (feature extractor + linear regressor), when the parameters of linear regressor are fixed and only the feature extractor need be trained, a discriminative extractor can still be obtained. In MIL models, the requirement for the feature is still linear separability. In the case of multiple branches, it becomes a joint optimization problem, and fixing the parameters of the regressor makes more easily training without causing performance degradation. Fixing the parameters of the linear regressor does not affect overall performance, but the different random initialization strategies have a slight impact on performance. In RGMIL, we adopt the kaiming_normal_() random initialization for the parameters of the linear regressor (with a mean of 0).
>
> Comment 8: Question-4
>
> [R] We added experiments on general multi-class problems (SIVAL and new MMNIST series) that don’t have form of max-based ranking, and details are given in global rebuttal.
>
> In the general multi-class scenario, during training, all bits in the indicator vector are considered reliable, and all corresponding branches participate in the calculation of loss function. If multiple branches output 1 during test, the branches can be sorted by querying the softmax high-bit values of each branch as confidence values, and the branch with the highest value can be selected as the final output. While in the new experiments, we did not perform extra processing on the outputs. We only modified the loss function during the training phase. Theoretically, RGMIL is flexible and can be fine-tuned based on specific circumstances.
>
> Comment-9: Limitation
>
> [R] Thanks, we will revise our description and add the analyses about the limitation. The contribution and additional experiments are shown in global rebuttal. All results presented are guaranteed to be reproduced on github.
>
> [1]R. Rahmani, et al., Localized content based image retrieval, in ACM SIGMM workshop, 2005.
>
> [2] J. Early, et al., Model Agnostic Interpretability for Multiple Instance Learning, in ICLR, 2022.

---

> > ### Comment · Reviewer_A2Ve · 2023-08-17
> >
> > Thank you for your rebuttal and extended results.
> >
> > * For SIVAL, why were 2, 4, and 10 positive class configurations considered, but not the full 25 class problem? This makes it difficult to compare to existing works that have used the complete configuration of this data.
> > * In general, I am unsure why only instance-level results are given for some datasets (e.g., MMNIST) and only bag-level results for others (e.g., SIVAL).
> >
> > While I understand the method better due to the rebuttal, I am still slightly unconvinced by the results (due to the problems outlined above).

---

> > > ### Author Response · Authors · 2023-08-18
> > > **Comments on new experiments**
> > >
> > > Thanks for your comment again.
> > >
> > > Comment-1: SIVAL results
> > >
> > > [R]1. Revised result table
> > >
> > > Our intention was to compare different pooling methods from binary classification to 11-class classification, with varying numbers of positive classes (1, 3, and 10). However, in Table 2 of the PDF in global rebuttal, we mistakenly filled in 2, 4, and 10.
> > >
> > > 2.Why not instance level?
> > >
> > > As an image retrieval problem, SIVAL consists of 25 classes of complex objects(including apple,book,goldmedal,...) s. We argue that SIVAL dataset is not quite suitable to be evaluated on single instance for following reasons:
> > >
> > > -  “fake” critical instances: In SIVAL , every bag is a complete image. The instances(image segments) in a bag are labeled positive if it contains target object, otherwise negative. But it’s not quite reasonable to expect correct predictions to be made for individual instances(image segments). For example, there are two bags (images), where one is an red apple image and another is the red book. Between two images, there are some similar instances (image segments), such as the center red regions of two images. If the instance-level is applied, though the center red region of the bag (the apple image) is labeled as positive during training, it is not sufficient to determine whether it belongs to a red apple or a red book when we make instance-level prediction in test.
> > >
> > > -  Instance dependency or orderings: Since every bag is a complete image, there are order or dependency relationships between image segments. And the set of several instances (image segments) represent one object, and not a single one can be omitted, such as an apple image (one image segment does not represent an apple, mentioned above). It means that the order or dependency is crucial and useful for the final target prediction. If instance-level is applied, that dependency will be missed. Thus it’s not really reasonable to predict an individual image segment.
> > >
> > > 3.Why not 25-classification?
> > >
> > >  - In the citation you mentioned above, J. Early, et al.[1] used 12 of 25 classes as positive classes. In our experiments, performance on a 11-classification task is similar to 13-classification. We considered the similar experiments series and presented the performance of different pooling methods. Apology for the misunderstanding that we need to provide evaluation of different pooling methods in different scenarios, but not a direct benchmark comparison. Since multiple MIL methods were evaluated on a 13-class classification problem in J's work:
> > >
> > > | Methods   | MI-Net | mi-Net | MI-Attn | MI-GNN |
> > > | --------- | ------ | ------ | ------- | ------ |
> > > | Model Acc | 0.819  | 0.808  | 0.813   | 0.781  |
> > >
> > > We can also provide performance evaluation for RGMIL in the 13-class scenario:
> > >
> > > | Methods    | MXP   | ABP   | G-ABP | DSP | RGP |
> > > | ---------- | ----- | ----- | ----- | ----- | ----- |
> > > | Model Acc  | 0.604 | 0.787 | 0.806 | 0.767 | 0.793 |
> > >
> > > The performance in the 25-class scenario is as follows:
> > >
> > > | Methods | MXP   | ABP | G-ABP | DSP   | RGP |
> > > | ------- | ----- | ----- | ----- | ----- | ----- |
> > > | Model Acc | 0.356 | 0.738 | 0.745 | 0.711 | 0.727 |
> > >
> > > Since our aim is to make comparison with other methods, in the experiments of 13 and 25-classification, we pick the branch with the highest confidence value to be the final output as we mentioned above in Comment-8 before, and we report the average of 10-time test accuracy  for each model after training convergence.
> > >
> > > Also notice that RGP and ABP assume no dependencies between instances, and any individual instance can be used to make predictions clearly. This differs from the actual data of SIVAL, where RGP and ABP series encounter similar situations and do not outperform many existing methods.
> > >
> > >
> > > Comment-2:
> > >
> > > R: Compared with other pooling methods, RGMIL explicitly enhances instance-level performance by transferring the learning task completely to the instance feature extraction stage. Since the primary focus of our MMNIST study was on instance-level performance analysis, we did not provide bag-level analysis and comparison in the paper. However, as you suggested, we have added the following comparison of bag-level performance for MMNIST general 4-class problems:
> > >
> > > | Aggregators | 10 / 10| 16 / 16| 32 / 32 | 64 / 64 | 512 / 512 |
> > > | ----------- | --------- | --------- | ------- | --------- | --------- |
> > > | MXP        | 43.46 | 40.70 | 60.56| 74.64 | 88.16 |
> > > | ABP         | 84.66 | **87.41** | **92.35**| 97.06 | 99.96|
> > > | G-ABP     | **85.20** | 87.22 |92.18| **97.46** | **1.0**|
> > > | DSP         | 84.07 | 83.62 | 90.17 | 95.89 | 99.98 |
> > > | RGP         | 82.70 | 87.26 | 89.80 | 97.22| 99.96 |
> > >
> > > The average of 5-time test accuracy is reported for each model after convergence or 50 epochs.
> > >
> > > We reiterate that any result in the paper can be reproduced with our codes.
> > >
> > > [1] J. Early, et al., Model Agnostic Interpretability for Multiple Instance Learning, in ICLR, 2022.

---

> > > > ### Comment · Reviewer_A2Ve · 2023-08-21
> > > >
> > > > Thank you for the additional comments and results. Apologies for my confusion on the SIVAL dataset - I did not realise the results in [1] only focused on a problem with 12 positive classes.
> > > >
> > > > Regarding the instance-level evaluation on SIVAL - the issue with image segments being difficult to separate makes some sense, but I do not see that as sufficient justification for not providing instance level results. First, this problem may not be limited to just this dataset. Second, all models have to deal with this same problem, therefore it is still worth comparing the relative performance on this dataset. Indeed, providing results and a discussion on the potential issues with the SIVAL dataset would enhance the work.
> > > >
> > > > In my opinion, the additional results do not show significant improvement for bag-level or instance-level prediction. As such, I am concerned about the general applicability of RGP and its performance relative to other approaches. I welcome further clarification of the merits of RGP over other methods, but at the moment will remain with my original score.
> > > >
> > > > Thank you again for your continued work on this!
> > > >
> > > > [1] J. Early, et al., Model Agnostic Interpretability for Multiple Instance Learning, in ICLR, 2022.

---

> > > > > ### Author Response · Authors · 2023-08-21
> > > > > **More analysis on newest experiments and  RGP**
> > > > >
> > > > > Thanks a lot for your comments. According to your comments, we also realize that when the object is much smaller than the background and the object can be represented by an or several instances in a bag (image blocks in an image), instance-level evaluation is still suitable. For this reason, the performance on the instance-level is also significant for demonstrating the performance of our method. Although the original motivation of designing our method is thought from the view of video sequence, we will try to test the instance-level performance on the SIVAL dataset. Due to time limitation, we did not finish this experiment at this moment. We will continue this experiment.
> > > > >
> > > > > For the problem of bag-level performances in added experiments, we make some newest analyzes and observations for this.
> > > > >
> > > > > Initially, RGP is designed for solving instance-level predictions in MIL scenario, and we have validated the advantage of RGP on instance level through both theoretical and experimental results. According to your comments, we found that its bag-level performance is not excellent like to instance-level. Based on this, we analyze the reason may be the effect of Eq.7. When we make bag-level predictions, Eq.7 brings some side effects: the weights of negative (non-critical) instances we obtained usually cannot be reduced to an extremely low level. The distribution of weights has been smoothed by the function.
> > > > >
> > > > > Despite providing consistency between instance-level prediction and bag-level training, this side effect limits the improvement in bag-level prediction performance for RGP.
> > > > >
> > > > > Compared with RGP, the parametric methods like ABP series could easily learn towards the direction with more extreme distributions--With the training process going, the ABP series methods intend to ignore more non-critical instances. Although the shared regressor scheme enhances the instance-level discriminability of RGP, the relatively smooth weights during bag-level prediction result in the mixture of more non-critical instances in the final bag-level representation, which brings some difficulties to completely migrate instance-level advantages to bag-level performance.
> > > > >
> > > > > As we shown in Table. 3 in paper, the distribution of weights we gained from RGMIL is smoothed:
> > > > >
> > > > > | Branch | 0 | 1 | 3 | 2 | 0 | 2 | 2 | 1 | 1 | 3 | $\mathbf{\hat{Y}}$ |
> > > > > | ------ | - | - | - | - | - | - | - | - | - | - | ---------------- |
> > > > > | 3      | 0.0151 | 0.0351 | $\mathbf{0.4981}$ | 0.1183 | 0.0220 | 0.0357 | 0.0365 | 0.0601 | 0.0313 | $\mathbf{0.1497}$ | 1 |
> > > > > | 2      | 0.0132 | 0.0577 | 0.0841 | $\mathbf{0.3062}$ | 0.0185 | $\mathbf{0.1959}$ | $\mathbf{0.1977}$ | 0.0299 | 0.069 | 0.0360 | 1 |
> > > > > | 1      | 0.0375 | $\mathbf{0.1889}$ | 0.0879 | 0.0479 | 0.0401 | 0.0793 | 0.0889 | $\mathbf{0.2001}$ | $\mathbf{0.1732}$ | 0.0563 | 1 |
> > > > >
> > > > > We also supplement the ABP information table(64/10) on MMNIST here:
> > > > >
> > > > > | Branch | 0| 1| 2| 0| 3| 3| 1| 0| 1| 3 | $\mathbf{\hat{Y}}$ |
> > > > > | ------ | - | - | - | - | - | - | - | - | - | - | ---------------- |
> > > > > | 3      | 0.0039| 0.0008| 0.0216| 0.0406| **0.2703**| **0.1102**| 0.0010| 0.0113| 0.0340|**0.5062** | 1 |
> > > > > | 2      | 0.0106| 0.0117| **0.3197**| 0.0555| 0.1789| 0.1574| 0.0126| 0.0084| 0.0442|0.2010 | 1 |
> > > > > | 1      | 0.0150| **0.3244**| 0.0985| 0.0090| 0.0324| 0.0072| **0.2214**| 0.0041| **0.1739**|0.1141 | 1 |
> > > > >
> > > > > In short, RGP provides a new interpretation and understanding of the MlL problem. RGP explicitly enhances instance-level performance by transferring the learning task completely to the instance feature extraction stage through a special aggregation method.
> > > > >
> > > > > For RGP, when we need instance-level predictions, the RGP performs significantly better. Combined with the analysis in section 4.3 and 4.4, we can see that with the faster convergence of $\phi$ to 0, RGP works more like a supervised algorithm on instance level, thus brings us more discriminative instance representations; with lower value of $\gamma$, RGP grasp critical instances more fairly.
> > > > >
> > > > > For ABP series, despite not emphasizing on instance-level representations, and even not like fully supervised algorithms, the more extreme weight distribution of RGP ensures that it assigns fewer weights to non-critical instances. This also contributes to its bag-level performance. In theory, RGP may not necessarily maintain its performance advantage on instance level in terms of bag-level performance.
> > > > >
> > > > > In Table 3 of global rebuttal, our additional experiments still validate the superiority of RGP in terms of instance-level. We are still willing to provide more instance-level evaluations to further validate the advantages of RGP. However, due to time constraints, we are currently unable to provide more experimental results.
> > > > >
> > > > > Thanks again for your review and all comments. We think these comments are helpful for improving our manuscript, and we also try our best to demonstrate our work better. We look forward to your recognition of our work.

---

### Official Review · Reviewer_oDjC · 2023-07-06

**Soundness:** 3 good
**Presentation:** 3 good
**Contribution:** 3 good
**Rating:** 6
**Confidence:** 4

**Summary:**

This paper tries to overcome the drawback of low instance-level prediction performance  in  the existing Multiple Instance Learning (MIL) models. According to the authors, the low instance-level performance is because the existing techniques focus mostly on analyzing the relationship between instances and aggregating them while ignoring learning of effective instance-level representation. To this end, this paper proposes a novel aggregator function called Regressor-Guided Pooling (RGP) that directly learns discriminative instance-level representation by mimicking human inference process. An extensive evaluation is performed using the multiple MIL datasets along with the pain estimation datasets to validate the effectiveness of the proposed aggregator function.

**Strengths:**

1. This paper has made a novel contribution to MIL field by identifying and addressing  key limitations of the proposed MIL techniques. The significance of the novel contribution is demonstrated through non-trivial performance gain  over other competitive baselines.
2. The motivation for introducing the Regressor-Guided Multiple Instance Learning (RGMIL) framework is very novel, natural, and intuitive.
3. This paper deals with the multi-classification scenario under the MIL setting which is unique and different from most of the MIL works that focus mostly on the binary-classification scenario. This unique scenario also helped to enhance the novelty of this paper.
4. The extensive evaluation is conducted considering multiple MIL datasets. In addition to the quantitative result, the authors have done great job explaining why the proosed technique works and how it avoids the limitations of the existing MIL techniques.

**Weaknesses:**

1. Most of the real-world video anomaly datasets are also used as a crucial testbed for the evaluation of MIL models [1, 2, 3].  Specifically, under video anomaly detection task, the MIL model is trained with video-level annotations (without having explicit access to the frame-level annotation). During the prediction process, the trained MIL model is used to perform frame-level prediction. The evaluation of the proposed MIL model on video-anomaly detection task is missing. I wonder how effective the proposed technique is in terms of making frame-level predictions. Also, in the related work section, the authors may need to explain how MIL techniques aimed to solve the video anomaly detection task are different from their work.
2. Some of the figures deserve better treatment. For example, the caption in Figure 1 is not very descriptive. Further,  modules in the figures (with texts) are difficult to read and understand.
3. The intuition behind Equation 6 is difficult to understand and is not very clear.


References:
[1]. Sultani et al. “Real-world Anomaly Detection in Surveillance Videos”. CVPR2018.
[2]. Sapkota et al. “Distributionally Robust Optimization for Deep Kernel Multiple Instance Learning”. AISTATS2021.
[3]. Tian et al. “Weakly-Supervised Video Anomaly Detection With Robust Temporal Feature Magnitude Learning”. ICCV2021.


**Questions:**

1. How effective the proposed technique is in the video-anomaly detection tasks compared to competitive baselines?

**Limitations:**

See weaknesses

---

> ### Author Rebuttal · Authors · 2023-08-09
>
>
> Thank you a lot for appreciating our efforts in addressing the use of multiple-instance learning (MIL) for instance-level predictions.
>
> Comment-1: Weaknesses (about applications)
>
> [R] In our method, RGP enhances the instance-level performance explicitly by transferring the learning task to the instance feature extraction stage through a special and simple aggregation method. This is theoretically proven in Section 4.3 of the main paper and Section A.1 of the supplementary material. RGP does not conflict with the current MIL-based video anomaly detection models. With the involvement of the instance-level feature extractor in training, RGP can theoretically achieve better instance-level performance than the existing ABP series aggregators. This work emphasizes generality and is applicable not only to pain estimation but also to other multi-classification problems. In the field of video anomaly detection, we can directly apply RGMIL for anomaly detection or simply replace the existing aggregators in other models with RGP to ensure instance-level performance.
>
> Besides, there are some similarities and differences worth noting between our experiments and video anomaly detection problems.
>
> Main similarities:
>
> - Both are image classification problems, may related to multi class;
>
> - Both emphasize instance-level representation;
>
>  Main Differences:
>
> - Our experimental scenario is related to the maximum label, but video anomaly detection typically does not involve the concept of a maximum label. Even in the case of multi-classification, there is no inherent ordering between class labels.
>
> - Some video anomaly detection datasets may provide videos that contain a large number of frames, potentially reaching thousands. Training single-frame features on such large videos may lead to insufficient GPU memory.
>
> Comment-2: Weaknesses (about figures)
>
> [R] Thanks a lot for your suggestion. We will revise these figures and descriptions so that they are more readable and easily understood.
>
> Comment-3: Weaknesses (about Equation 6)
>
> As a non-parametric dynamic aggregate component, RGP still share the same view with attention mechanism that weights should rely on the feature itself. It is also reasonable to order the instances’ importance by their classification score that indicates the likelihood of being positive (critical instances) and assign weights accordingly. We could process the current instance classification score provided by regressor in different ways. We tested different combinations of regression methods and weight acquisition methods:
>
> - the output score is in the form of 1-dimensional logits output with a sigmoid function, and the weights are obtained based on the logits output;
>
> - the output score is in the form of 2-dimensional logits output with a softmax function, and the weights are obtained based on the high-bit output of logits;
>
> - the output score is in the form of 2-dimensional logits output with a softmax function, and the weights are obtained by dividing the high-bit output of logits by the low-bit output;
>
> - the output score is in the form of 2-dimensional logits output with a softmax function, and the weights are obtained by subtracting the low-bit output of logits from the high-bit output.
>
> These designs have the same underlying principles but differ in real numerical computations. We found that the last method had significantly better performance, and then we used and formulated it as Equation 6 in this paper. In practice, the sigmoid output barely works. But as shown in section A.2 of the supplementary material, there are other methods that also works without Equation 6.
>
> The method of normalization also affects the performance, and we tested parameterized normalization and unnormalized methods, both of which were not as good as simple unparameterized normalization. The two equations are only training tricks that make RGP more practical.
>
> Comment 4: Questions
>
> Thank you a lot for your suggestions. Due to the time limitations, we did not add this experiment on the dataset of video anomaly detection in this version, but we add experiments on two general multi-instance multi-class image datasets: SIVAL[1] and the new MMNIST series, as shown in the PDF in global rebuttal. Anomaly detection is not our major research interests, but we are also interested in the results for video anomaly detection problems. The experiments on anomaly detection will be very meaningful, and we will try to test it later.
>
>  [1] Rouhollah Rahmani, et al. "Localized content based image retrieval." Proceedings of the 7th ACM SIGMM international workshop on Multimedia information retrieval. 2005.

---

> > ### Comment · Reviewer_oDjC · 2023-08-18
> > **Comments on rebuttal**
> >
> > Thank you for  clarifying Equation 6. Currently, I do not have additional questions. I would love to see incorporation of our discussion into the revised paper.

---

### Official Review · Reviewer_mChi · 2023-07-06

**Soundness:** 3 good
**Presentation:** 3 good
**Contribution:** 3 good
**Rating:** 7
**Confidence:** 4

**Summary:**

This paper presents a new model, the Regressor-Guided Multiple Instance Learning network (RGMIL), which addresses the challenge of providing discriminative instance-level representation in multi-classification MIL scenarios. It introduces an aggregator, the Regressor-Guided Pooling (RGP), that enhances the instance-level performance of Multiple Instance Learning (MIL) tasks. RGMIL outperforms existing methods in various datasets and demonstrates the potential for instance-level predictions.

**Strengths:**

1.  The paper presents a novel model, the Regressor-Guided Multiple Instance Learning network (RGMIL), which brings a fresh perspective to the field of multi-classification MIL scenarios.
2. The paper is well-structured and the problem and proposed solution are explained clearly.
3. The experiments are well-designed and the promising results effectively support the authors' claims.
4. The paper is inspiring for future research, including potential applications for multi-classification tasks such as the UCF-Crime dataset.

**Weaknesses:**

1. The paper lacks a clear definition and purpose for the metrics used in Table 4. Additionally, a thorough analysis of the results presented in the table is missing.

**Questions:**

See the weaknesses.

**Limitations:**

See the weaknesses.

---

> ### Author Rebuttal · Authors · 2023-08-09
>
>  Thank you a lot for appreciating our efforts in addressing the use of multiple-instance learning (MIL) for instance-level predictions.
>
> Comment-1: Weaknesses
>
> [R]The detailed information of the metrics can be found in section C in supplementary. We will give more details on the UNBC experiments and the additional information for Table 4. All the results regarding MMNIST and benchmark tests in the paper can be reproduced within 3 hours. Besides, the result on UNBC can also be reproduced in days.
>
> Additionally, we provided supplementary results on two general multi-instance multi-class image datasets: SIVAL[1] and the new MMNIST series experiments, as shown in the PDF in global rebuttal. We next try to test on image classification datasets next, like video anomaly datasets and more general multi-class scenarios.
>
> [1]Rahmani, Rouhollah, et al. "Localized content based image retrieval." Proceedings of the 7th ACM SIGMM international workshop on Multimedia information retrieval. 2005.

---

> > ### Comment · Reviewer_mChi · 2023-08-18
> >
> > Thank you for clarifying the availability of metric definitions in Supplementary Section C. However, I'm interested in understanding how each metric evaluates specific aspects of the model and the corresponding detailed analysis of experimental results under each metric.

---

> > > ### Author Response · Authors · 2023-08-19
> > > **Comments on evaluation metrics**
> > >
> > > Thank you for your comment. We provide the explanation of metrics as follows:
> > >
> > > MAE (Mean Absolute Error), which can be seen as L1 loss, is a commonly used loss function for regression models. It is the sum of the absolute differences between the target values and the predicted values.
> > >
> > > MSE (Mean Squared Error), which can be seen as L2 loss, is also a widely used regression loss function. It is the sum of the squared differences between the predicted values and the true values. By squaring the errors (letting $\varepsilon$ = true value - predicted value), MSE amplifies the errors if $\varepsilon$  > 1. If there are outliers in the data, the value of $\varepsilon$  can be large, and thus $\varepsilon$ squared will be much larger than |$\varepsilon$|. Therefore, compared to using MAE to compute the loss, using MSE gives more weight to the outliers.
> > >
> > > We use these two metrics to measure the overall error between the predicted values and the ground truth at the frame level during test. Lower values of MAE and MSE indicate more accurate predictions by the model. In the UNBC experiments, due to the imbalanced distribution of pain levels, a large number of frames have a pain level of 0. Therefore, if accuracy metric is used for model evaluation, it can easily overestimate the model performance. Hence, accuracy metric is generally not used for model evaluation.
> > >
> > > PCC(Pearson Correlation Coefficient) measures the linear correlation between two variables, typically the predicted values and the ground truth. It ranges from -1 to 1, where a value close to 1 indicates a strong positive linear correlation, a value close to -1 indicates a strong negative linear correlation, and a value close to 0 indicates no linear correlation. PCC is widely used to assess the overall relationship or agreement between two continuous variables.
> > >
> > > ICC (Intraclass Correlation Coefficient) is a statistical measure that quantifies the consistency or reliability of measurements made by different observers or raters. It is commonly used when there are multiple raters or multiple measurements taken on the same subjects. ICC ranges from 0 to 1, where a value close to 1 indicates high agreement or consistency among the raters or measurements, and a value close to 0 indicates low agreement.
> > >
> > > MAE (Mean Absolute Error) and MSE (Mean Squared Error) are loss functions that quantify the error or discrepancy between predicted and true values. They are used during model training and optimization. Lower values of MAE and MSE indicate better model performance.
> > > PCC and ICC, on the other hand, are evaluation metrics that assess the agreement, correlation, or consistency between predicted and true values. They provide insights into the quality of predictions and the reliability of measurements. Higher values of PCC and ICC indicate better agreement or consistency.
> > >
> > > Most of the supervised models use some of the above metrics to evaluate performance on pain estimation problems. With the sampe experimental settings, our model demonstrates similar performance to fully supervised models on these metrics, indicating comparable reliability in pain estimation tasks.

---

> > > > ### Comment · Reviewer_mChi · 2023-08-19
> > > >
> > > > Thank you for your thorough explanation of the metrics used in your study. I appreciate your attention to addressing the concerns raised. I look forward to seeing the revised and enhanced version of your paper.

---

### Official Review · Reviewer_4ybk · 2023-07-27

**Soundness:** 3 good
**Presentation:** 2 fair
**Contribution:** 3 good
**Rating:** 5
**Confidence:** 3

**Summary:**

This paper introduces _Regressor-guided MIL network (RGMIL)_ as a solution to address the challenges in Multiple instance learning, particularly in the multi-class classification scenario for pain-estimation. RGMIL introduces a novel aggregator, Regressor-Guided Pooling (RGP) in place of currently widely used attention-based aggregators. The design of RGP is based on simulating the correct inference process of humans when facing similar problems. The experiments show RGP out-performing evaluation benchmarks in MIL and works similar to supervised models for pain estimation.

**Strengths:**

- The aggregator introduced in this work is grounded in prior knowledge of Multiple Instance Learning task construction. As a result, rather than opting for a complex black box approach, the design remains simplistic.This work aligns with recent research trends that emphasize leveraging simple yet valuable prior knowledge and inductive biases to design more effective solutions.
- Regressor-Guided Pooling (RGP), outperforms all MIL evaluation benchmarks including SOTA by a large margin (Table 1) and also works on-par with supervised models for pain estimation (Table 4). This compelling result clearly highlights the potential of the introduced method.
- The code available has a well-written README and looks easy to reproduce.

**Weaknesses:**

- **Missing dataset details**: The details of dataset(s) construction for MIL in pain-estimation setting is missing. One of the contributions mentioned in this work is RGMIL being the first weakly supervised deep model for pain estimation, so I feel it is important for this work to also focus on pain estimation datasets and benchmarks.
    - Firstly, it is not clearly mentioned whether the results presented in section 4.5 is for UNBC pain dataset. The way the dataset is constructed for the MIL setting is also not presented which makes it difficult to understand if the dataset is different or the same as supervised setting.
    - Secondly, the result is presented on only a single dataset. Given that the contributions of the work also focuses on being the first weakly supervised model for pain estimation, it is important to benchmark on more datasets (for example BioVid dataset presented in [25]).


- **Issues in paper structure and writing:** The paper lacks a cohesive flow, making it difficult to follow. The writing style is inconsistent both within the sections and throughout the entire document. This paper needs major restructuring and rewriting to be easy to follow and understand.
    - The main paper contains many non-essential details and lacks conciseness. Few suggestions improve it: the assumptions section (3.1) should be condensed into smaller paragraphs, with more extensive details provided in the supplementary material. Similarly, sections 4.2, 4.3, and 4.4 are verbose and could benefit by retaining crucial details in the main paper and moving the remaining content to the supplementary.
    - In its current stage, some important contributions are left behind in supplementary (A1, A2) which supports the method’s effectiveness.
    - Figure and table captions (except Table 1) are not self-sufficient and needs referencing to the text for understanding. Authors should revise the captions for better stand-alone clarity.
    - Figure 1(b) is not clearly visible or understandable.


- **Missing important citations and explanations:** There are important statements without citations or explanations, based on which the paper makes assumptions for model design (lines 143-144) and presents results on a different dataset (lines 230-231).
    - line 143-144: 'Experience has shown that when we add the aggregator ⇢ to the backbone + regressor model for MIL problems, the instance-level performance often decreases significantly.' I would suggest the authors to cite works that show that adding aggregator decreases instance-level performance. If it is by observation across paper, the authors should present this in a table or a figure to give strong basis for assumptions.
    - line 230-231: 'Considering multiple factors, the UNBC pain dataset isn’t really a clear and flexible enough choice to present a demonstration.' What are the multiple factors here is not clear and I would suggest the authors to explain clearly.



- **Little to no focus on second contribution:** The paper's focus on the first contribution is appreciated, but it leaves little room to highlight the second contribution, leading to confusion in understanding the relevance of pain estimation in this work. It was referenced multiple times but only given a limited space (10 lines: 326-336) in the overall paper. To improve clarity, the authors can choose to implement either of these two suggestions:
    - Prioritize RGP as the main contribution and dedicate the paper to thoroughly highlighting its significance, while briefly mentioning pain estimation as a valuable benefit of the method only shortly. This also relieves the authors of having to benchmark any other datasets while fully highlighting the method and its effectiveness.
    - Condense some verbose sections, to be more concise, and move extra details to supplementary. This ensures that the authors have more space to present the benchmark results on pain estimation.

**Questions:**

- My main suggestions for the authors is to address the writing in the paper and to highlight their contributions accordingly. The results looks very compelling, but the paper is quite difficult to follow to appreciate the design or importance of the method.
- Other minor changes include: captions for standalone clarity, dataset details for reproducibility and clear understanding.

**Limitations:**

The authors have not mentioned any possible limitations or potential negative societal impact of this work. I would suggest the authors to include some details in settings where this method may not be useful.

---

> ### Author Rebuttal · Authors · 2023-08-09
>
> Thank you for appreciating the effectiveness and novelty of our approach.
>
> Comment-1: About writing and descriptions.
>
> [R] Thanks for your suggestions, and we will revise our writing so that the proposed method can be easily understood.
>
> Comment-2：Missing dataset details
>
> [R] We construct the UNBC dataset for MIL as follows: we selected consecutive 64 frames in the video as a training bag. To increase the amount of data, we used a sliding window approach with a step size of 8 to produce training bags. Overall, each frame was sampled as an instance in different bags 8 times. In each fold of the experiment, the total number of training bags in the constructed MIL dataset averaged around 6000. Detail will be added into the article.
>
> In experiments of pain estimation, only UNBC dataset is used, because we only obtain this one public dataset. For BioVid, before, we have sent our application for downloading this data, but not receiving any replies. At present we have another pain dataset collected by Xi Jing Hospital, but this cannot be made public until now.
>
> Additionally, we supplemented experiments on SIVAL[1] and new MMNIST for general multi-class classification, shown in PDF of global rebuttal.
>
> Comment-3：about line 143-144
>
> [R] Compared with the fully supervised mode, the available information obtained by MIL is limited. Therefore, the limit of the instance-level performance of MIL should be equivalent to the fully supervised mode. Two main architectures of MIL, introduced in section 2.1, differ from the fully supervised mode only on the presence of the aggregator. We think that the inability of MIL to achieve the performance of the fully supervised mode is due to the limitation of the learning stage of the new component (aggregator). This can be verified by the results of MMNIST presented in Table 2. As seen in the first column in Table 2, when we implement experiments with the current mainstream models under the mode 10/1 (train/test bag size), the performance is obviously worse than 1/1 (fully supervised) mode. This illustrates that the current aggregators are unable to obtain enough accurate information for the aggregation from the 10 instances, and it does not surpass the information obtained from a single instance in the fully supervised mode. This leads to inadequate performance.
>
>
> Comment-4：about line 230-231
>
> [R]
>
> - The UNBC dataset is of extremely class imbalance, and it’s not normal for MIL problem. For UNBC, the four metrics presented in Table 4 are usually used to validate the performance, since the accuracy in the UNBC dataset is easily inflated and is not reliable due to the imbalance distribution of data. Whereas, accuracy is the most intuitive and reliable metric, which we strongly prefer to use.
>
> - In experiments of UNBC dataset, 25 folds cross-validation is widely used to test the performance. It takes over a week to obtain one overall evaluation result, which reduces flexibility during the demonstration process and prevents conducting comparative experiments with multiple aggregators under different modes.
>
> - Both single-frame image and the feature extractor itself are large, and during training, increasing the bag length easily leads to insufficient GPU memory, limiting the experimental scenarios.
>
> Comment-5: Little to no focus on second contribution
>
> [R]  Thanks for your suggestion. We agree with your concern that the contributions we presented is indeed not clear enough. Different from the mainstream MIL methods, RGMIL addresses the general multi-class classification problem in MIL scenarios by directly learning discriminative instance-level features.
>
> Initially, we try to design a pain estimation method from the view of MIL. In studying process, we find that it places strong emphasis on single-frame predictions during testing for pain estimation. However, most of existing MIL methods focus more on learning the aggregator black box itself, rather than learning single-frame features, meanwhile ignoring the instance-level performance. We also found that most MIL methods mainly focused on the binary-classification problem, little considering the multi-classification problem.
>
> Motivated by limitations of existing MIL methods, we designed RGMIL that explicitly enhances instance-level performance by transferring the learning task completely to the instance feature extraction stage through a special aggregation method. It is theoretically proved (in section 4.3 of paper and section A.1 of supplementary) that it is beneficial to improve instance-level performance. RGMIL emphasizes generality, and it is not used only on pain estimation but also other multi-classification problems. In our work, we validate the performance on benchmark datasets, MMNIST series, and a real application (the pain dataset, UNBC).
>
> Comment-6: Limitation
>
> [R]Feasibility of training at instance-level with excessively large bag-lengths: RGMIL significantly improves instance-level performance than current methods. However, as described in section A.1 of supplementary, we think that MIL problems may have ambiguity inherently in extreme scenarios. It isn’t theoretically insured whether training instance feature extraction directly is feasible in extreme cases. As shown in Table 1, with a maximum of bag length 512, instance performance decays with increasing bag lengths. Due to the memory limitation, we did not achieve the experiment on longer bag-lengths (like several thousands) to confirm the feasibility. Thereafter, we will try our best to figure it out.
>
> Memory limitation: It is challenging to handle the excessively large bags when the feature extractor is involved in the training. For example, in the MMNIST experiment (with a single bag of length 1024), there is insufficient memory on an NVIDIA 4090 graphics card.
>
> [1]Rahmani, Rouhollah, et al. "Localized content based image retrieval." Proceedings of the 7th ACM SIGMM international workshop on Multimedia information retrieval. 2005.

---

> > ### Comment · Reviewer_4ybk · 2023-08-14
> > **Follow-up comments**
> >
> > I appreciate the authors' response in clarifying all my questions and comments. I hope that they will do their due diligence when submitting the final version. I have additional replies to authors' responses listed below:
> >
> > (1) Given the paper writing was difficult to follow due to verbosity and lack of self-explanatory figure captions, I strongly suggest to pay attention in rewriting this work for the final version. Especially, since the results are promising, a good writing will help in understanding the importance of the proposed method.
> >
> > (5) Thanks for explaining your process to design the RGMIL pipeline. I agree with the statement stating your contributions: "In our work, we validate the performance on benchmark datasets, MMNIST series, and a real application (the pain dataset, UNBC)." Please clearly rewrite the contributions in the paper to reflect this.
> >
> > (2),(3),(4),(6)  please include the following points in the revised final version:
> > - Additional dataset details in the paper for completeness.
> > - For lines 143-144 and 230-231, add an additional 1-2 sentence short explanation to the paper to include what was explained in your reply to my questions.
> > - Also add limitations mentioned here in the final version.

---

> > > ### Author Response · Authors · 2023-08-16
> > > **Comments on current revisions**
> > >
> > > Thanks for your suggestion again. According to your suggestions, we have finished partial revises, and some details are shown in the following part. We will continue to revise our writing.
> > >
> > > More clear distribution declaration:
> > >
> > > 1.In Introduction, we modify the first sentence of second distribution as: “The effectiveness of RGMIL is validated in pain estimation.”
> > >
> > > 2. Specifically, in Conclusion, we have added: “We also validated RGMIL in real application on UNBC dataset”. after prioritizing the idea of RGP as the main contribution.
> > >
> > > Clarifying ambiguous figures and tables:
> > >
> > > 1.We have redrawn the model diagram in Fig. 1 to ensure clear visibility. Fig. 2 has also been redrawn, with axis labels added to each subfigure.
> > >
> > > 2.The titles of all figures and tables have been modified to ensure clarity. For example, the new title of table 3 is “RGP Information Table (64/10) on MMNIST. The first row contains the 10 images of the test bag. The Indicator vector $\mathbf{\hat{Y}}$ is the model output. The remaining contents are all the weights of corresponding instances from all the branches”.
> > >
> > > Some important explanations:
> > >
> > > 1.about line 143-144:
> > >
> > > The new descriptions are shown as following: ‘Compared with supervised learning, the available information obtained by MIL is limited. Therefore, the limit of the instance-level performance of MIL should be equivalent to the fully supervised mode. Two main architectures introduced above differ from the fully supervised ones only on the presence of the aggregator. We argue that the inability of MIL to achieve the performance of the fully supervised mode is due to the limitation of the learning stage of the new component (aggregator).”
> > >
> > > 2.about line 230-231:
> > >
> > > The new description is shown as following: ‘Due to the extremely imbalance of pain intensity distribution, the pain dataset is not really a clear and flexible enough choice to present a demonstration.’
> > >
> > > 3.The way is to construct UNBC for MIL dataset:
> > >
> > > We supplement this part in the title of Table in which the UNBC results are shown. The new title has been modified to “Instance-level Performance Comparison with Supervised Models (*: S-O-T-A model). In the video of UNBC dataset, consecutive 64 frames are regarded as a training bag. To increase the amount of data, we use a sliding window approach with a step size of 8 to produce the training bags. The first row contains the four evaluation metrics. Following the same experimental setting with MSRAN, this experiment is implemented via 25-fold cross-validation, and the result is the mean of 25-fold cross-validation result. Statistics collected from Deep Pain, DSHF, DBR, Multistream CNN, MSRAN, LIAN” for stand-alone clarity.
> > >
> > > More clarity in theoretical analysis:
> > >
> > > 1.Section 3.1 has been amended to briefer paragraphs.
> > >
> > > 2.Reduce the verbosity of sections 4.2, 4.3, and 4.4. For example, line 290-295 was deleted.
> > >
> > > 3. Additionally, important analysis left behind in supplementary (A1, A2)  including the ablation study of RGP has been moved to section 4.4 and a newly added section 4.5 .
> > >
> > > More Experiments and analysis:
> > >
> > > 1.Additional general multi-classification experiments (SIVAL and new MNIST series) provided in rebuttal phase have been supplemented.
> > >
> > > 2. Besides, the 2 major limitations mentioned above have been added in the newly added section 4.7 of the main paper.

---

> > > > ### Comment · Reviewer_4ybk · 2023-08-17
> > > > **Thanks for the response**
> > > >
> > > > Thanks for mentioning some of the changes in the paper in the new response. I have no more questions, and I hope this discussion can be useful when preparing a revised version.

---

### Author Rebuttal · Authors · 2023-08-09

About RGMIL:

RGMIL can address the general multi-class classification problem in MIL scenarios by learning discriminative instance-level features. It holds a different view that we can solve MIL problems by learning on instance level like human does. Inherently coming with interpretability, RGP significantly enhanced instance-level performance by transferring the learning task to the instance feature extraction stage through a special and simple aggregation method. The advantage is theoretically proven in Section 4.3 of the main paper and Section A.1 of supplementary material. When there is emphasis on single-instance predictions, RGP can theoretically achieve better instance-level performance than the existing aggregators. This work emphasizes generality and is applicable not only to pain estimation but also to other general multi-classification problems.

About Equation 6:

As a non-parametric dynamic aggregate component, RGP still share the same view with attention mechanism that weights should rely on the feature itself. It is also reasonable to order the instances’ importance by their classification score that indicates the likelihood of being positive (critical instances) and assign weights accordingly. We could process the current instance classification score provided by regressor in different ways. We tested different combinations of regression methods and weight acquisition methods:

1.the output score is in the form of 1-dimensional logits output with a sigmoid function, and the weights are obtained based on the logits output;

2.the output score is in the form of 2-dimensional logits output with a softmax function, and the weights are obtained based on the high-bit output of logits;

3.the output score is in the form of 2-dimensional logits output with a softmax function, and the weights are obtained by dividing the high-bit output of logits by the low-bit output;

4.the output score is in the form of 2-dimensional logits output with a softmax function, and the weights are obtained by subtracting the low-bit output of logits from the high-bit output.

These designs have the same underlying principles but differ in real numerical computations. We found that the last method had significantly better performance, and then we used and formulated it as Eq.6 in this paper. In practice, the sigmoid output barely works. But as shown in section A.2 of supplementary material, there are other methods that also works without Eq.6.

The method of normalization also affects the performance, and we tested parameterized normalization and unnormalized methods, both of which were not as good as simple unparameterized normalization. The two equations are only training tricks that make RGP more practical.

About supplementary experiments：

As requested, we additionally conducted experiments on two general image multi-classification datasets as supplements in the PDF:

- Table 1 and Figure 1 in the PDF are the MMNIST series instance-level results in the main paper. According to suggestions, we improved the presentation of charts and titles. In this series of experiments, for the MMNIST dataset, the label is determined by the maximum value in the input bag, and the model's output during test is obtained by selecting the branch with the highest position among all branches with an output of 1.

- Table 2 is the bag-level performance reports on SIVAL dataset. SIVAL consists of 25 classes of complex objects photographed in different environments, where each class contains 60 images. Each image is segmented into approximately 30 segments, and each segment is represented by a 30-dimensional feature vector that encodes information, such as the segment’s colour and texture. The segments are labelled as containing the object or containing background. We selected 1,310 classes as positive classes, and randomly sampled from the other classes as the negative class. We set up three Linear+ReLU blocks as feature extractors. The test scenario involves image classification problems ranging from binary to 11-class classification. Each image is treated as a bag, resulting in bag-level performance evaluation. Considering that the classes in the SIVAL dataset do not have any ranking relationship, all bits of the output indicator vector are reliable, so all branches are involved to the computation of loss function. Continuing with the current version's approach, the model's output during test is still obtained by selecting the branch with the highest position among all branches with an output of 1. Despite learning instance-level features only, RGP still exhibits impressive performance on bag level.

- Table 3 is the instance-level performance reports on MMNIST. We adopted a different approach in constructing the dataset this time: while still using the 4-branch RGMIL, we selected images with labels ranging from 0 to 6 as the negative class (background class), and images with labels 7, 8, and 9 as three different positive classes. Similar to SIVAL, there is no sequential relationship among the different positive class labels. Each positive class and the negative class account for approximately 25% of the dataset. Within one positive bag, the number of images of positive class is approximately 10%. Like to the SIVAL tests, during training, all bits of the output indicator vector are reliable, so all branches are involved to the computation of loss function. Continuing with the current version's approach, the model's output during test is still obtained by selecting the branch with the highest position among all branches with an output of 1. Unlike the original MMNIST series, the convergence speed of different methods varies much. For example, it's very difficult for MAX and DSP to converge even training for hundreds of epochs. Thus, we present the average of 10-time test accuracy is taken for each model after convergence or training after 100 epochs. RGP still shows dominance in the scenario on instance level.

---

> ### Comment · Area_Chair_CKBG · 2023-08-18
> **Further discussions warranted?**
>
> Dear authors, reviewers,
>
> I am glad to see the ongoing discussion between authors and reviewers, I greatly welcome the response of the other reviewers as well (either as a reponse to this message or to the rebuttal of your review), as this will help will reaching a concensus.
>
> AC

---

### Decision · Program_Chairs · 2023-09-21

**Decision:**

Accept (poster)

**Comment:**

This paper received three positive ratings (5,6,7) and one negative rating (3). Given the lack of consensus amongst the reviewers, the AC examined the (shared) concerns. Overall, the reviewers find the paper interesting and with strong performance. The main open concern as expressed by the negative reviewer is the general applicability of the problem in MIL. The authors have provided extra instance- and bag-level experiments. The instance-level results are promising, while competitive results are obtained for the bag-level experiments. Since the method is mostly designed for instance-level performance, the AC deems that the authors have sufficiently addressed the concerns of the reviewer. The AC therefore recommends acceptance.